



Atmospheric
Measurement
Techniques

# Discriminating between clouds and aerosols in the CALIOP version 4.1 data products

**Zhaoyan Liu**[1], **Jayanta Kar**[1,2], **Shan Zeng**[1,2], **Jason Tackett**[1,2], **Mark Vaughan**[1], **Melody Avery**[1], **Jacques Pelon**[3], **Brian Getzewich**[1,2], **Kam-Pui Lee**[1,2], **Brian Magill**[1,2], **Ali Omar**[1], **Patricia Lucker**[1,2], **Charles Trepte**[1], and **David Winker**[1]

[1]NASA Langley Research Center, Hampton, VA, USA
[2]Science Systems and Applications (SSAI), Hampton, VA, USA
[3]LATMOS, Sorbonne Université, Université de Versailles Saint Quentin, CNRS, Paris, France

**Correspondence:** Zhaoyan Liu (zhaoyan.liu@nasa.gov)

**Abstract.** CE1 CE2 The Cloud-Aerosol Lidar and Infrared Pathfinder Satellite Operations (CALIPSO) mission released version 4.1 (V4) of their lidar level 2 cloud and aerosol data products in November 2016. These new products were derived from the CALIPSO V4 lidar level 1 data, in which the calibration of the measured backscatter data at both 532 and 1064 nm was significantly improved. This paper describes updates to the V4 level 2 cloud–aerosol discrimination (CAD) algorithm that more accurately differentiate between clouds and aerosols throughout the Earth's atmosphere. The level 2 data products are improved with new CAD probability density functions (PDFs) that were developed to accommodate extensive calibration changes in the level 1 data. To enable more reliable identification of aerosol layers lofted into the upper troposphere and lower stratosphere, the CAD training dataset used in the earlier data releases was expanded to include stratospheric layers and representative examples of volcanic aerosol layers. The generic "stratospheric layer" classification reported in previous versions has been eliminated in V4, and cloud–aerosol classification is now performed on all layers detected everywhere from the surface to 30 km. Cloud–aerosol classification has been further extended to layers detected at single-shot resolution, which were previously classified by default as clouds. In this paper, we describe the underlying rationale used in constructing the V4 PDFs and assess the performance of the V4 CAD algorithm in the troposphere and stratosphere. Previous misclassifications of lofted dust and smoke in the troposphere have been largely improved, and volcanic aerosol layers and aerosol layers in the stratosphere are now being properly classified. CAD performance for single-shot layer detections is also evaluated. Most of the single-shot layers classified as aerosol occur within the dust belt, as may be expected. Due to changes in the 532 nm calibration coefficients, the V4 feature finder detects ~ 9.0 % more features at night and ~ 2.5 % more during the day. These features are typically weakly scattering and classified about equally as clouds and aerosols. For those tropospheric layers detected in both V3 and V4, the CAD classifications of more than 95 % of all cloud and daytime aerosol layers remain unchanged, as do the classifications of ~ 89 % of nighttime aerosol layers. Overall, the nighttime net cloud and aerosol fractions remain unchanged from V3 to V4, but the daytime net aerosol fraction is increased by about 2 % and the daytime net cloud fraction is decreased by about 2 %.

## 1 Introduction

The Cloud-Aerosol Lidar and Infrared Pathfinder Satellite Operations (CALIPSO) mission has provided unique height-resolved measurements of aerosols and clouds on a global scale since 2006 (Winker et al., 2010). These data have been used in a wide variety of studies of phenomena such as intercontinental dust transport, cloud microphysics, and ocean ecosystems, which are documented in numerous publications (e.g., Z. Liu et al., 2008a; D. Liu et al., 2008; Huang et al., 2008; Chand et al., 2009; Uno et al., 2009; Solomon

et al., 2011; Vernier et al., 2013; Yu et al., 2015; Ma et al., 2015; Cesana and Waliser, 2016; Jing et al., 2016; Tan et al., 2016; Behrenfeld et al., 2017). The cloud–aerosol discrimination (CAD) algorithm uses CALIPSO backscatter measurements and retrieved spatial properties to separate clouds from aerosols and must perform reliably under a wide variety of conditions to deliver the necessary information for additional level 2 lidar data processing and support the widest possible range of scientific investigations.

The primary payload aboard CALIPSO is the Cloud Aerosol Lidar with Orthogonal Polarization (CALIOP). The CALIOP laser emits pulses of linearly polarized light at 532 and 1064 nm and separately measures the backscattered laser energy polarized parallel ($\parallel$) and perpendicular ($\perp$) to the transmitted beam at 532 nm and the total backscattered energy at 1064 nm (Hunt et al., 2009). The nighttime 532 nm measurements are calibrated using the molecular normalization technique at stratospheric altitudes (Powell et al., 2009; Kar et al., 2018), and this nighttime calibration is the foundation for subsequent calibration of the daytime 532 nm data and all 1064 nm data. The recently released CALIOP V4 level 1 data include major modifications to the calibration algorithms at both 532 and 1064 nm that produce substantially more accurate profiles of attenuated backscatter coefficients at both wavelengths (Getzewich et al., 2018; Kar et al., 2018; Vaughan et al., 2019).

In the CALIOP data processing sequence (Winker et al., 2009), the calibrated level 1 data are first analyzed using an adaptive thresholding scheme to detect layer boundaries at single shot (333 m), 1, 5, 20, and 80 km horizontal averaging resolutions (Vaughan et al., 2009). Layers detected at finer resolutions generally are denser (i.e., have larger backscatter) than layers detected at coarser resolutions. The next step is to determine if the detected layers are clouds or aerosol layers. This is achieved through the CAD algorithm that uses multidimensional probability density functions (PDFs) derived from an extensive training set of CALIOP measurements to accurately distinguish clouds from aerosol layers (Liu et al., 2004, 2009, 2010). The CAD classifications are in turn used as primary inputs to two further classification algorithms: the CALIOP aerosol subtyping algorithm, which now identifies different aerosol species in both the troposphere (Omar et al., 2009) and stratosphere (Kim et al., 2018), and the CALIOP ice–water phase algorithm, which uses layer-integrated attenuated backscatter and layer-integrated volume depolarization ratio to discriminate between ice clouds and water clouds (Hu et al., 2009; Avery et al., 2018). Optical depths and profiles of particulate backscatter and extinction coefficients are then retrieved from the fully classified layers using a suite of hybrid extinction retrieval algorithms (Young and Vaughan, 2009; Young et al., 2013, 2018).

The CAD PDFs constructed for the initial release of the CALIPSO data products used three dimensions, the layer-mean attenuated backscatter at 532 nm, $\langle \beta'_{532} \rangle$, the layer-mean total attenuated color ratio, $\chi' = \langle \beta'_{1064} \rangle / \langle \beta'_{532} \rangle$, and

the mid-layer altitude of the detected features, $z_{\mathrm{mid}}$ (Liu et al., 2009). Additional dimensions of layer-mean 532 nm volume depolarization ratio, $\delta_{\mathrm{v}} = \langle \beta'_{532,\perp} \rangle / \langle \beta'_{532,\parallel} \rangle$, and latitude were added to the CAD PDFs that were subsequently used in the version 3 (V3) data products released in May 2010 (Liu et al., 2010). The addition of $\delta_{\mathrm{v}}$ has significantly improved the classification of dense dust layers that were frequently misclassified as cloud over dust source regions in the version 1 and version 2 data releases (e.g., Chen et al., 2010). However, there remained some instances of dense dust near the source regions and transported dust at high altitudes that were misclassified in the V3 data releases (Jin et al., 2014). This is due partly to the PDFs, which were not fully optimized for dust identification, and partly to the algorithm design, which required that all layers detected at single-shot resolution be classified as clouds by default without applying the CAD algorithm to them. Other scenarios that were persistently misclassified in the prior data releases were smoke layers at high altitudes and fresh volcanic aerosol layers in the upper troposphere and lower stratosphere.

The release of the fully recalibrated V4 level 1 data required substantial updates to the five-dimensional (5-D) set of PDF parameters previously used in the V3 level 2 analyses. Accordingly, an entirely new set of PDFs was constructed and subsequently used to process the V4 level 2 data. Apart from using an extended training set to develop the V4 PDFs, several structural changes were also made. These include a finer latitudinal resolution than in V3, as well as extending the altitude range to stratospheric altitudes. This enabled the application of these PDFs to volcanic layers and the occasional cloud and smoke layers detected in the lower and mid-stratosphere.

In yet another important application, the new CAD algorithm is now applied to all layers detected at single-shot resolution, even though these layers were not used in the training sets used for building the PDFs themselves. The single-shot layer detection scheme is applied to the 1064 nm attenuated backscatter measurements between the surface and $\sim 8.2$ km (Vaughan et al., 2009) and is specifically designed to identify only the densest, most strongly scattering features present in the CALIOP measurements. While the layers detected at single-shot resolution are predominantly clouds, there are occasions when very dense aerosol masses are seen embedded within large-scale dust storms, smoke plumes, and/or marine layers. Correctly classifying these features as aerosols, rather than clouds, is essential for accurately characterizing the upper range of aerosol extinction coefficients that occur on the planet.

In this paper we provide a comprehensive description of the numerous improvements made to the CALIOP CAD algorithms. We first provide the motivation for developing the new PDFs in Sect. 2 and then describe the development of the new PDFs in Sect. 3. In Sect. 4 we present the overall differences between V3 and V4 5 km layer classifications, followed by a more complete range bin-by-range bin as-

sessment of the performance of the new CAD algorithm in Sect. 5. The assessment is carried out in the troposphere and in the stratosphere, including at polar altitudes and for layers detected at single-shot resolution. We present the performance of single-shot classification in Sect. 6. We additionally describe a set of post-processor algorithms designed to handle several generic cases that are not well classified using the PDFs alone and thus require special consideration. Conclusions and a summary of all changes are given in Sect. 7.

## 2 Motivations for modifying the CAD algorithm

The V3 and V4 CAD algorithms are based on five different parameters ("dimensions"). One of the crucial dimensions is the total attenuated color ratio, $\chi'$, of the layer under consideration. The color ratio used in the CAD algorithm in turn depends upon the calibration of the attenuated backscatter coefficients in both channels. The V4 level 1 data incorporate significant improvements in the calibration of these channels, presented in several accompanying publications (Getzewich et al., 2018; Kar et al., 2018; Vaughan et al., 2019). In particular, the changes in the calibration of the 1064 nm channel have been substantial. Improved selection of calibration targets (i.e., more homogenous cirrus clouds), estimation of the 1064 nm calibration scale factor using multiple granules, and calculation of calibration coefficients as a function of granule elapsed time have all led to significant improvements in the 1064 nm backscatter data (Vaughan et al., 2019).

Figure 1 shows a comparison of the joint occurrence frequency of layer $\langle \beta'_{532} \rangle$ and $\chi'$ for all layers detected at 5 km horizontal averaging resolution between altitudes of 0 and 1 km in V3 (upper panels) and V4 (lower panels). The data are composited in 20° latitude bands extending from the Antarctic to 10° S. Features occupying this altitude range consist mainly of boundary layer aerosols (left-bottom corner clusters in each panel) and water clouds (right-upper corner clusters). While in the V4 data there appears to be only one mode for water clouds that is concentrated at roughly the same value (i.e., $\chi' \approx 1.2$ for each latitude band), the distribution of the water cloud cluster in the V3 data is latitudinally dependent and splits into two modes at latitudes south of 70° S (Fig. 1a). This is due mainly to the calibration of the V3 1064 nm data, for which a constant calibration scale factor was applied to the entire orbit to transfer the 532 nm calibration to the 1064 nm data (Vaughan et al., 2010). Using a constant scale factor fails to fully compensate for thermally induced intra-orbit variations in the calibration coefficients, which cause the scale factor to vary with latitude (Hunt et al., 2009; Vaughan et al., 2019). The CAD PDFs are built by fitting the joint distributions of $\langle \beta'_{532} \rangle$ and $\chi'$ as functions of $\delta_V$, latitude, and altitude, and thus the significant changes in the V4 data calibration illustrated by the example in Fig. 1 necessitate the generation of a new set of PDFs.

There have also been several problems noticed in the performance of the V3 CAD algorithm. For instance, dense aerosol (dust) layers over the Taklimakan Desert are sometimes misclassified as clouds in the V3 data products (Jin et al., 2014). Dust layers that are lofted from the Asian deserts and transported northward to the Siberian and American Arctic regions are often classified as ice clouds (Di Pierro et al., 2011, 2013; J. Huang et al., 2015). Also, smoke layers at high altitudes were occasionally misclassified as cirrus cloud by the V3 CAD algorithm (Miller et al., 2011; J. Huang et al., 2015). Correct classification of smoke layers is particularly affected by the strong differential absorption between the two channels linked to the presence of fine-mode carbonaceous particles. Taken together, these classification problems pointed to inadequacies in the CAD algorithm that needed reconsideration.

A total of 1 full year CE3 (2008) of the CALIOP 5 km layer product was used to develop and test the V3 PDFs. However, stratospheric features were excluded. In the previous releases, any feature detected in the stratosphere was flagged as a generic "stratospheric feature", and no further classifications were attempted. Part of the reason for not extending the earlier CAD algorithms into the stratosphere was that there are ubiquitous polar stratospheric clouds (PSCs) detected by CALIOP in the stratosphere during both polar winters, and prior to launch there was insufficient knowledge of the spectrally dependent backscatter from PSCs to reliably classify them. In the 12 years since the launch of CALIPSO, detailed studies have been performed to characterize PSCs based on the CALIOP measurements, and specialized algorithms have been developed to specifically identify different PSC types (Pitts et al., 2009, 2013). The exclusion of stratospheric features from the V3 and earlier CAD test data, together with the fact that there are fewer aerosols at higher altitudes, led to the lack of sufficient constraints to build accurate PDFs in the upper troposphere and stratosphere. As a result, V3 cloud and aerosol identification in the upper troposphere was not as reliable as at lower altitudes.

Over the years, the acquisition of more high-altitude aerosol measurements (for instance, from the eruption of several volcanoes that injected aerosol plumes into the stratosphere) and a better understanding of PSC optical and physical properties suggested that the original decision to exclude stratospheric layers from the CAD analysis could be successfully revisited. Therefore, when building the V4 PDFs, the 2008 test data were augmented with additional data from June 2011 and all stratospheric features were included. The Nabro and Puyehue-Cordón volcanos erupted in June 2011, and volcanic aerosol layers were observed in the upper troposphere and lower stratosphere in both the Northern Hemisphere and Southern Hemisphere (Fairlie et al., 2014; Fromm et al., 2014; Kim et al., 2018). Adding the June 2011 data thus provides the stratospheric aerosol observations needed for comprehensive PDF generation.

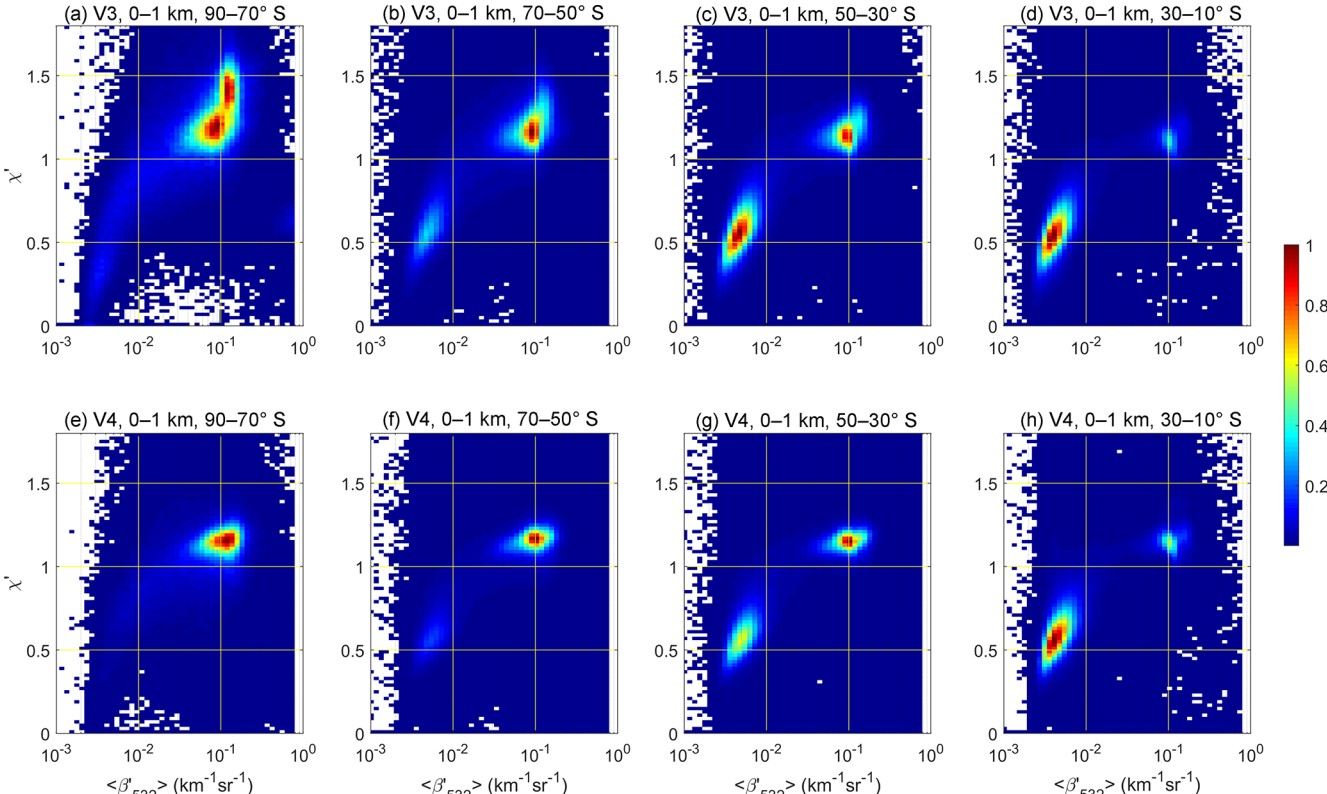

**Figure 1.** Distributions of occurrence frequency normalized by the maximum occurrence as a function of total color ratio ($\chi'$) and mean attenuated backscatter ($\langle \beta'_{532} \rangle$) for all layers detected at 0–1 km altitudes within latitude bands of 90 [TS1] to 70° S **(a, e)**, 70 to 50° S **(b, f)**, 50 to 30° S **(c, g)**, and 30 to 10° S **(d, h)**, using data from all of 2008 and June 2011. V3 distributions are shown in the upper panels; V4 distributions are shown in the lower panels.

In V3 and previous versions, CAD was not applied to layers detected at single-shot resolution. Based largely on 50 h of data acquired during the Lidar In-space Technology Experiment (LITE; Winker et al., 1996), prelaunch expectations for CALIOP were that the maximum backscatter coefficients in dense aerosol layers would be too small to be reliably detected at single-shot resolution. Consequently, all layers detected at single-shot resolution were classified by default as clouds, with no CAD analysis being necessary. Increased analysis and understanding of the data indicated that some of the layers observed to be fully embedded within more extended plumes of dust and smoke and detected at single-shot resolution are likely to be legitimate aerosol layers and thus should be evaluated by the CAD algorithm. This desire to apply the CAD algorithm to all detected layers, coupled with the significantly improved calibrations, led to the reworking of the CAD PDFs described in the next section.

## 3 The V4 CAD algorithm

### 3.1 Building V4 PDFs

The V3 CAD algorithm is based on the following confidence function (Liu et al., 2010):

$$f = \tag{1}$$
$$\frac{P_c\left(\langle \beta'_{532} \rangle, \chi', \delta_v, z_{\text{mid}}, \text{lat}\right) - P_a\left(\langle \beta'_{532} \rangle, \chi', \delta_v, z_{\text{mid}}, \text{lat}\right)}{P_c\left(\langle \beta'_{532} \rangle, \chi', \delta_v, z_{\text{mid}}, \text{lat}\right) + P_a\left(\langle \beta'_{532} \rangle, \chi', \delta_v, z_{\text{mid}}, \text{lat}\right)}.$$

In this equation, $P_c$ and $P_a$ are the 5-D PDFs for cloud and aerosol, respectively. $z_{\text{mid}}$ is the mid-layer altitude, and lat is the layer latitude. The function $f$ is a normalized differential probability that ranges from $-1$ to $1$. The CAD score reported in the CALIOP level 2 products converts $f$ to a percentile (integer) ranging from $-100$ to $100$. A feature is classified as cloud when $f \geq 0$ and as aerosol when $f < 0$. The absolute value of the CAD score provides a confidence level for the classification.

In the construction of V4 PDFs, the training dataset for a given altitude range (0–1, 1–2, ..., 7–8, 8–10, 10–12, 12–16, and 16–25 km) and latitude band (every 10° from 90° S to 90° N) is sliced into 10 subsets based on the $\delta_v$ (i.e., $< 3\%$,

3 %–6 %, 6 %–10 %, 10 %–15 %, 15 %–20 %, 20 %–25 %, 25 %–30 %, 30 %–35 %, 35 %–40 %, and > 40 %). To simplify the PDF construction, two-dimensional (2-D) Gaussian functions are used to represent the distributions of clouds and aerosols in the $\chi' - \langle\beta'_{532}\rangle$ plane for each $\delta_v$ slice using

$$
p_s = A \exp \left\{ - \left[ a_s \left( \ln\langle\beta'_{532}\rangle - \ln\langle\beta'_{532}\rangle_{s,0} \right)^2 \right. \right. \tag{2}
$$
$$
+ 2b_s \left( \ln\langle\beta'_{532}\rangle - \ln\langle\beta'_{532}\rangle_{s,0} \right) \left( \chi' - \chi_{s,0} \right)
$$
$$
\left. \left. + c_s \left( \chi' - \chi_{s,0} \right)^2 \right] \right\},
$$

where $a_s = \frac{\cos^2\theta_s}{2\sigma^2_{\langle\beta'_{532}\rangle s}} + \frac{\sin^2\theta_s}{2\sigma^2_{\chi'_s}}$, $b_s = -\frac{\cos 2\theta_s}{4\sigma^2_{\langle\beta'_{532}\rangle s}} + \frac{\sin 2\theta_s}{4\sigma^2_{\chi'_s}}$, and $c_s = \frac{\sin^2\theta_s}{2\sigma^2_{\langle\beta'_{532}\rangle s}} + \frac{\cos^2\theta_s}{2\sigma^2_{\chi'_s}}$. $A_s$ is a scaling factor ($0 \le A_s \le 1$) that determines the overall occurrence probability for a cluster of species $s$, where $s = $ a, i, or w, where a indicates aerosol, i indicates ice clouds, and w indicates water clouds. $\langle\beta'_{532}\rangle_{s,0}$ and $\chi'_{s,0}$ represent the characteristic scattering properties of a cluster, $\sigma^2_x$ is the in-cluster variance of quantity $x$ (e.g., of $\ln(\langle\beta'_{532}\rangle)$ and $\chi'$), and $\theta_s$ is the orientation angle of the cluster. The construction of the CAD PDFs determines a set of characteristic PDF parameters $A_s$, $\langle\beta'_{532}\rangle_{s,0}$, $\chi'_{s,0}$, $\sigma^2_{\langle\beta'_{532}\rangle s}$, $\sigma^2_{\chi'_s}$, and $\theta_s$ for each given latitude, altitude, and $\delta_v$ range based on the CALIOP measurement data. $A_s$, $\langle\beta'_{532}\rangle_{s,0}$, $\chi'_{s,0}$, $\sigma_{\langle\beta'_{532}\rangle s}$, and $\sigma_{\chi'_s}$ can be determined in the grids in which there is only one feature type (i.e., aerosol, water cloud, or ice cloud) or where there are multiple types of features that separate well (such as the $\delta_v > 6$ % cases shown in Fig. 2). Interpolation and/or extrapolation are then used to determine these parameters for the grids in which either the clusters do not separate well or there are not sufficient data for some feature types. After using the training dataset to determine a global set of these PDF parameters, the V4 PDFs are interpolated to a uniform size of 1 km for the altitude dimension from the surface to 18 km and 5° for the latitude dimension from 90° S to 90° N. Above 18 km, the PDFs retain their latitude dependence but are no longer altitude dependent.

Figure 2 shows the V4 PDFs (contours) derived from the V4 training data for the 1–2 km altitude range and 20 to 30° N latitude band. Because this latitude band and altitude range extend through the dust belt of the Northern Hemisphere (D. Liu et al., 2008), dust aerosols are ubiquitous. There are also other types of aerosol present (e.g., the cluster of points with $\delta_v < \sim 6$ % labeled as "Other" in Fig. 2k), such as maritime, continental, and smoke, or mixtures of these other types with some amount of dust. Dust aerosol generally has a large particulate depolarization ratio at 532 nm due to the irregular shape and relatively large size of dust particles and therefore can be easily identified from the CALIOP measurements (D. Liu et al., 2008; Z. Liu et al., 2008a, b, 2015; Omar et al., 2009). However, it is too warm for ice clouds to form within this altitude and latitude range, as shown in Figs. 1 and 2. In this case, $A_i$ for the ice PDF (Eq. 2) is zero or some

very small value. The other five ice cloud PDF parameters in this example are determined by interpolation or extrapolation from those at higher altitudes and latitudes.

Figure 3 shows the CALIOP V4 training data (colored 2-D distributions) along with the V4 $\langle\beta'_{532}\rangle_{a,0}$, $\langle\beta'_{532}\rangle_{i,0}$, and $\langle\beta'_{532}\rangle_{w,0}$ parameters (red, yellow, and green dashed lines, respectively) for the 20 to 30° N latitude band in all altitude ranges from the surface to 10 km to show the evolution of different clusters as the altitude changes. A clear distribution mode starts to appear for ice clouds at altitudes above 7 km. There is also a small fraction of ice clouds detected in the 6–7 km altitude range and there may be some ice clouds in the 5–6 km altitude range that are not clearly seen. The PDF parameters, $\langle\beta'_{532}\rangle_{s,0}$ and $\chi'_{s,0}$, for ice clouds can be determined more accurately at relatively high altitudes > 7 km in this latitude band because there are many more ice clouds detected at high altitudes. Extrapolation is used to derive these ice PDF parameters at low altitudes where almost no ice clouds are detected. Interpolation is used as required to determine the ice cloud PDF parameters at low latitudes from those at high latitudes. Meanwhile, dense and depolarizing dust clusters are clearly seen below 5 km and there may be some dust above 5 km. The PDF parameters can be determined more accurately below 5 km and extrapolation is used to determine the aerosol PDF parameters at high altitudes. By using this combination of extrapolation and interpolation, all the PDF parameters can be determined globally for all feature clusters in each grid cell in the $\delta_v - z - $ lat space. This technique helps fill the grid cells in which there are either no data or insufficient data for PDF construction.

Figure 4 presents another example of the V4 CALIOP training dataset (2008 and June 2011) and V4 PDFs constructed for a latitude band of 50 to 40° S and altitude range of 12–16 km. For comparison, Fig. 5 shows the V3 CALIOP training data (2008) and V3 PDFs for the same latitude band and altitude range as in Fig. 4. We note that there are almost no useful samples for small $\delta_v$ values (< 10 %) in either the V3 or V4 training datasets. Compared with the V3 training dataset, in which only one distribution mode is seen (Fig. 5), the V4 training dataset has two distribution modes for large $\delta_v$ values (Fig. 4e–j). By comparing the 2008 and June 2011 periods, we find that the upper mode corresponds to ice clouds, as seen in the V3 training dataset, and the lower mode corresponds to volcanic ash aerosol from the June 2011 Puyehue-Cordón volcanic eruption in southern Chile. The June 2011 data thus provide the stratospheric aerosol properties needed for comprehensive PDF generation. This relatively fresh volcanic aerosol has large 532 nm backscatter coefficients, similar to those of ice clouds, but a much smaller color ratio. In constructing the V4 aerosol PDFs, the characteristic values for aerosol $\langle\beta'_{532}\rangle_0$ range between $1.6 \times 10^{-3}$ and $3.8 \times 10^{-3}$ km$^{-1}$ sr$^{-1}$, while the characteristic values for $\chi'_0$ are $0.5 < \chi'_0 < 0.6$. In contrast, the V3 $\langle\beta'_{532}\rangle_0$ and $\chi'$ parameters for aerosol appear to be too small and not representative for this fresh volcanic aerosol, especially for large $\delta_v$

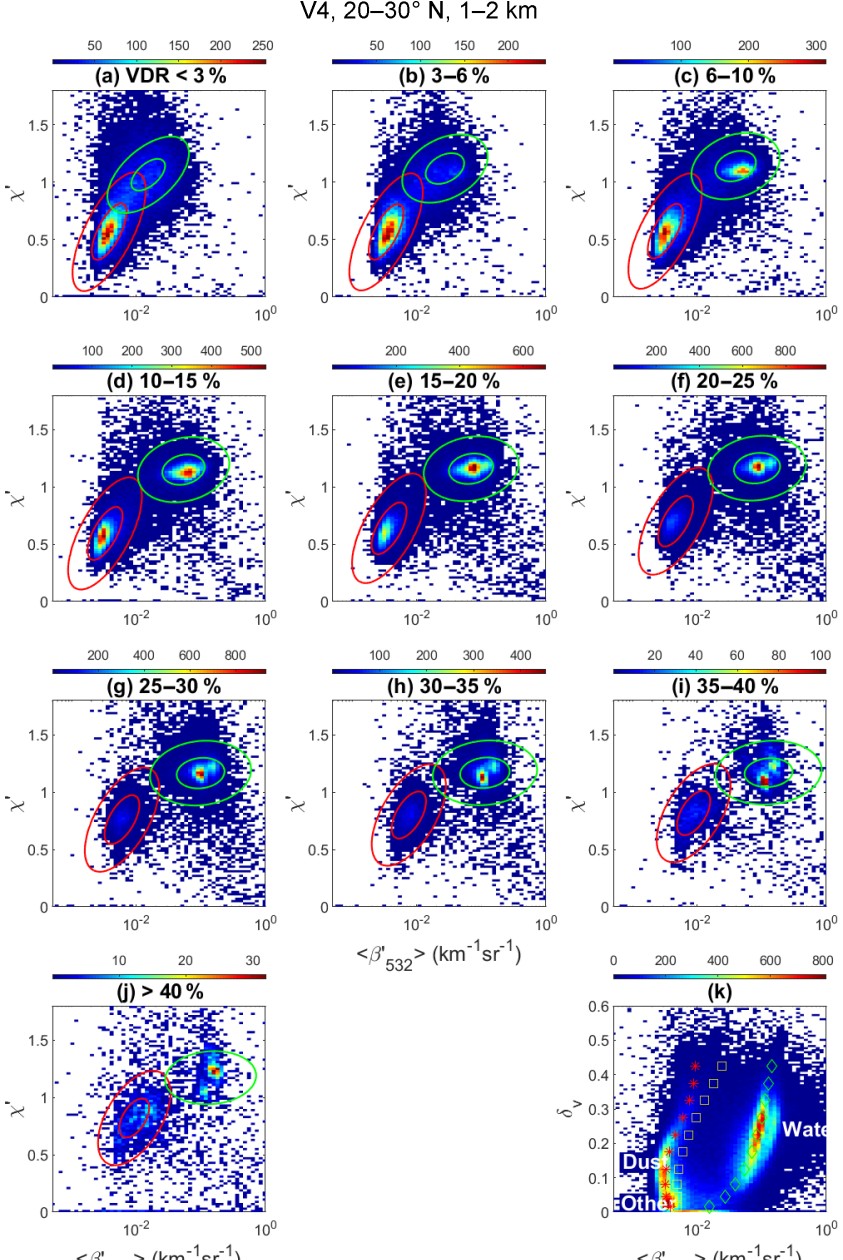

**Figure 2. (a–j)** V4 CALIOP measurement data (colored 2-D distributions) acquired during all of 2008 and June 2011, along with PDFs constructed for aerosols (red contours) and clouds (yellow contours) at two levels of 0.05 and 0.5 in the $\chi' - \langle \beta'_{532} \rangle$ space measured for a latitude band of 20 to 30° N and an altitude range of 1–2 km for the 10 depolarization grids, and **(k)** V4 data in the $\delta_v - \langle \beta'_{532} \rangle$ space along with the $\langle \beta'_{532} \rangle_0$ parameters used in constructing PDFs for aerosol (red asterisks), ice cloud (yellow squares), and water cloud (green diamonds). Note that, at this latitude band and altitude range, there are almost no ice clouds detected. Consequently, the scaling factor A for the ice PDF (Eq. 2) is nearly zero and the ice contours are not visible in panels **(a–j)**.

values; however, these V3 parameters may prove to be appropriate for aged volcanic aerosols or background aerosols (Jäger and Hofmann, 1991; Gobbi, 1995). Also, the in-cluster variance parameters ($\sigma^2_{\langle \beta'_{532} \rangle}$ and $\sigma^2_{\chi'}$) are smaller in V4 than in V3 as a result of the significant improvement in the level 1 data calibration (also see Fig. 1).

In the V3 PDFs, the scaling factor A for aerosols was set to 0 at high altitudes when $\delta_v > 0.03$ (i.e., any layers in the upper troposphere and lower stratosphere that had $\delta_v > 0.03$ were classified as cloud and assigned a CAD value of 100). This led to misclassifications of relatively fresh volcanic aerosols with high ash content as high confidence ice clouds.

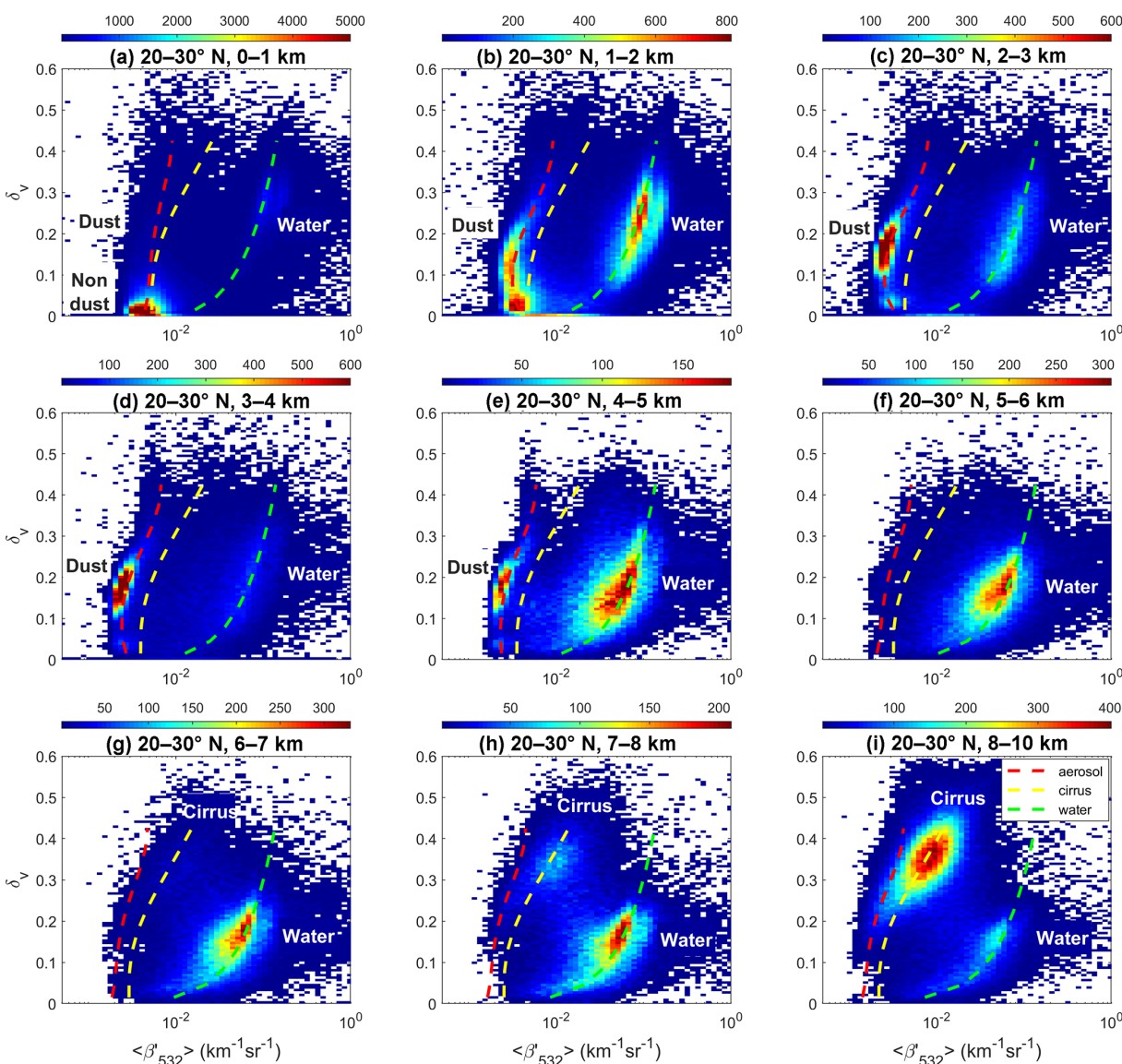

**Figure 3.** CALIOP measurements (colored 2-D distributions) and the $\langle \beta'_{532} \rangle_0$ parameters used in constructing PDFs for aerosols (red dashed line), ice clouds (yellow dashed line), and water clouds (green dashed line), showing the evolution of clusters with increasing altitude in the 20 to 30° N latitude band. The labels "non-dust", "dust", "ice", and "water" indicate, respectively, where non-dust aerosol or a mixture of non-dust aerosol with a small amount of dust aerosol, dust aerosol, ice cloud, and water clouds is clearly identifiable. In general, the water cloud cluster is largely separated from the aerosol clusters and is hence most easily discriminated from aerosols. Relatively significant amounts of dust can be seen up to 5 km and some dust can still be identified above 5 km in the 5–6 km altitude range in this latitude band. Ice clouds normally form at high altitudes and a significant amount of ice cloud is already seen in the 6–7 km altitude range. They can form at even lower altitudes, down to $\sim 4$ km for this latitude band (Campbell et al., 2015). The most difficult scenario to discriminate in this latitude band is the dust that is relatively dense and at altitudes above $\sim 6$ km where the ice cloud can occur frequently. There is an overlap between relatively dense dust and ice cloud at 5–7 km altitudes, although both of them occur at very infrequently. Moving north toward the Arctic, this overlap region moves to lower altitudes (not shown) because ice clouds tend to form at lower altitudes, whereas the occurrence frequency of relatively dense dust decreases quickly when moving poleward.

We note that the overall occurrence frequency of volcanic aerosols is very small, and the contribution of the misclassified aerosol at high altitudes to the overall misclassification rate is generally not significant. Conversely, because the occurrence frequency of cirrus clouds at high altitudes is quite large, most high cirrus were classified correctly as clouds. However, a large majority of the high clouds identified in V3 have CAD scores (i.e., confidence levels) of 100, which suggests that their classification confidence levels could be systematically overestimated.

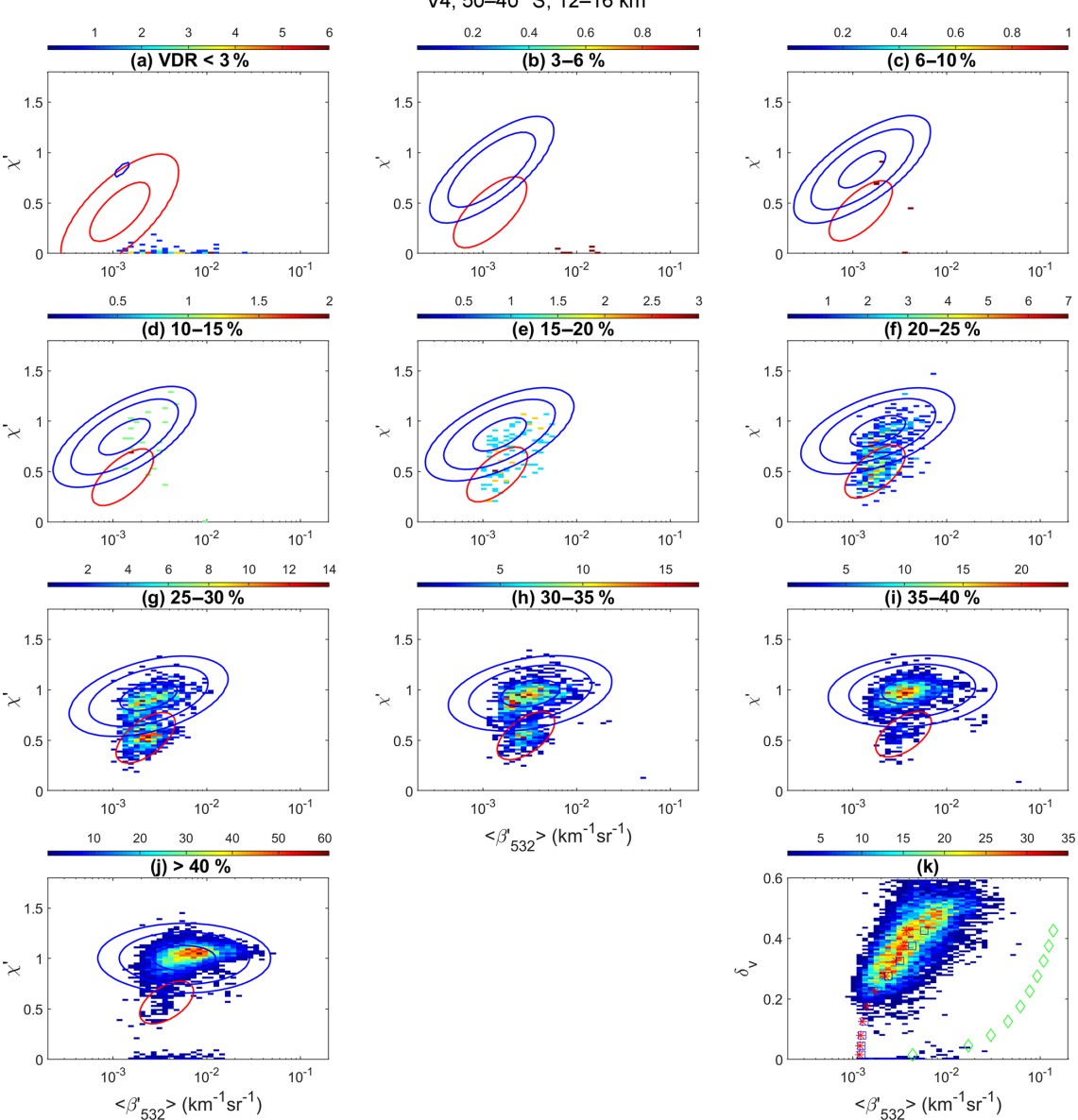

**Figure 4.** Panels **(a)**–**(j)** show the joint distributions of V4 CALIOP measurements of $\chi'$ and $\langle \beta'_{532} \rangle$ (colored 2-D distributions) for each of the PDF depolarization ratio intervals. These data were acquired during 2008 and June 2011 over a latitude band of 50 to 40° S and an altitude range of 12–16 km and were used to construct the V4 CAD PDFs that are applied within this same latitude–altitude region. Also shown in each panel are the derived PDFs for ice clouds (blue contours at three levels of 0.005, 0.05, and 0.5) and aerosols (red contour). Panel **(k)** aggregates all data in panels **(a)**–**(j)** and replots them CE4 in $\delta_v - \langle \beta'_{532} \rangle$ space, along with the $\langle \beta'_{532} \rangle_0$ values used to construct the PDFs for aerosols (red asterisks), ice clouds (blue squares), and water clouds (green diamonds).

## 3.2 CAD post-processor algorithms

After initial classification using the CAD PDFs, two additional algorithms are applied to mitigate two common errors. We introduce these algorithms in the following subsections.

### 3.2.1 "Fringe amelioration" via spatial proximity analysis

The 532 nm calibration coefficients in V4 are systematically lower than the V3 values by 3 % to 12 %, depending on latitude, season, and lighting conditions (Kar et al., 2018; Getzewich et al., 2018). These lower calibration coefficients increase the magnitude of the V4 532 nm attenuated backscatter coefficients, thereby facilitating the detection of optically

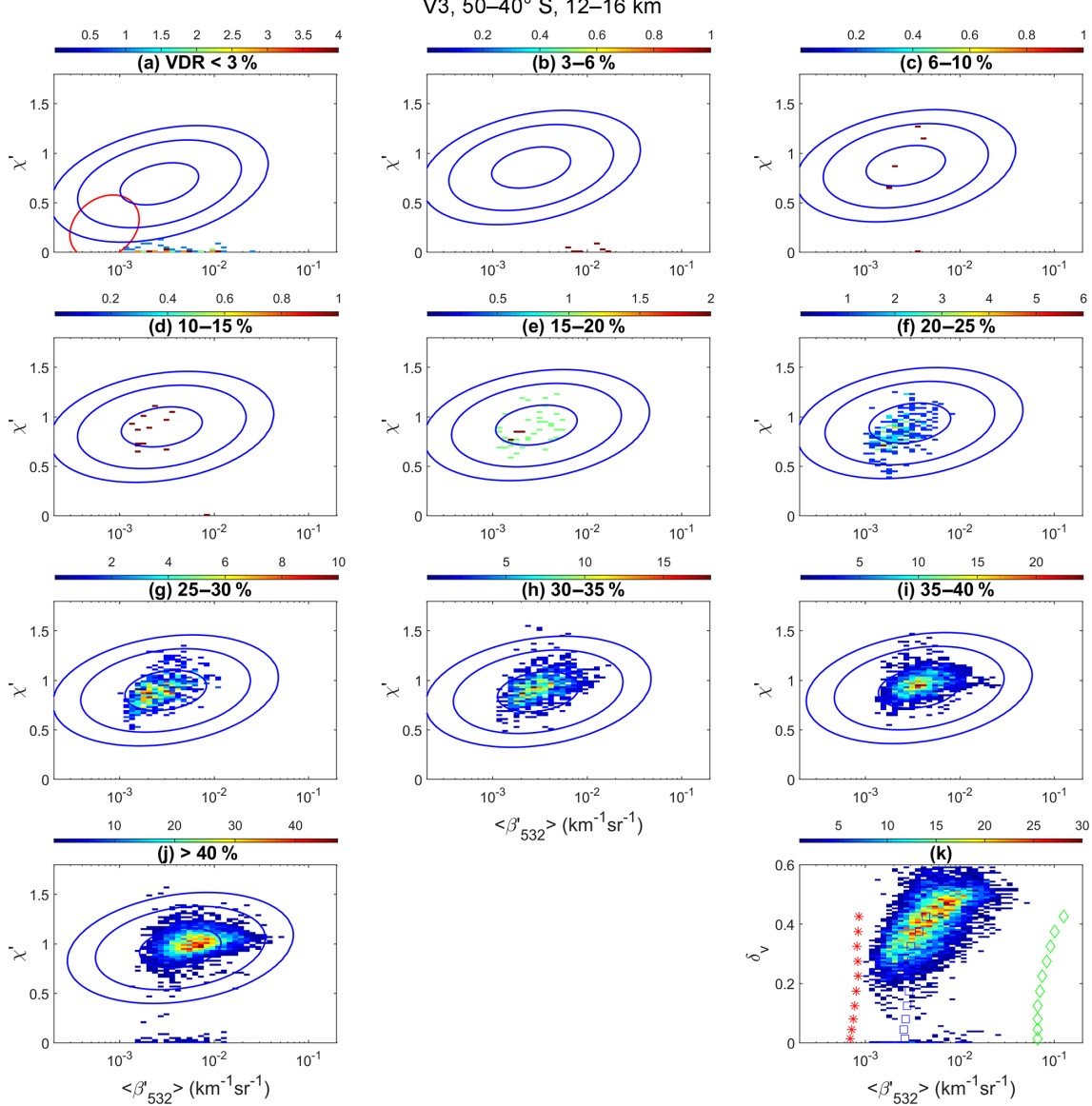

**Figure 5.** The same as in Fig. 4 but for V3 data acquired during 2008 only and used in the V3 PDF construction and the V3 PDFs (contours).

thinner layers than were previously detected in V3. One notable side effect of this improvement is the increased occurrence of weakly scattering features located along the edges of ice clouds. These features are detected at 20 or 80 km horizontal averaging resolutions, and, as illustrated in Fig. 6, occur at the horizontal edges and along the lower boundaries of more robust cirrus layers that are detected at 5 km resolution.

Because these features are, by definition, always found adjacent to cirrus clouds, they are referred to as "cirrus fringes", which are a new feature in the V4 dataset. As described earlier, 1 month of volcanic aerosol data were added to the 1-year training dataset. These additional training data helped better constrain the characteristic PDF parameters (especially $\langle \beta'_{532} \rangle_0$ and $\chi'_0$) at high altitudes, and the new

V4 aerosol PDFs are now more sensitive to lofted depolarizing aerosols at high altitudes. As illustrated in detail in Sect. 5, this increased sensitivity helps better differentiate lofted aerosol layers from clouds. At the same time, however, there is a cost to be paid for this increase in sensitivity, as the V4 CAD algorithm preferentially classifies cirrus fringes as depolarizing aerosols, not clouds.

Figure 7 shows joint histograms of $\chi'$ and $\log_{10}(\langle \beta'_{532} \rangle)$ for cirrus fringes and the adjacent cirrus clouds for data acquired between 40° N and 40° S during January, February, and December 2008. The relationship between the two cluster centroids is similar to the separation seen in Fig. 2; just as the aerosols in Fig. 2 exhibit distinctly lower $\chi'$ and $\log_{10}(\langle \beta'_{532} \rangle)$ values than the water clouds, the fringes in

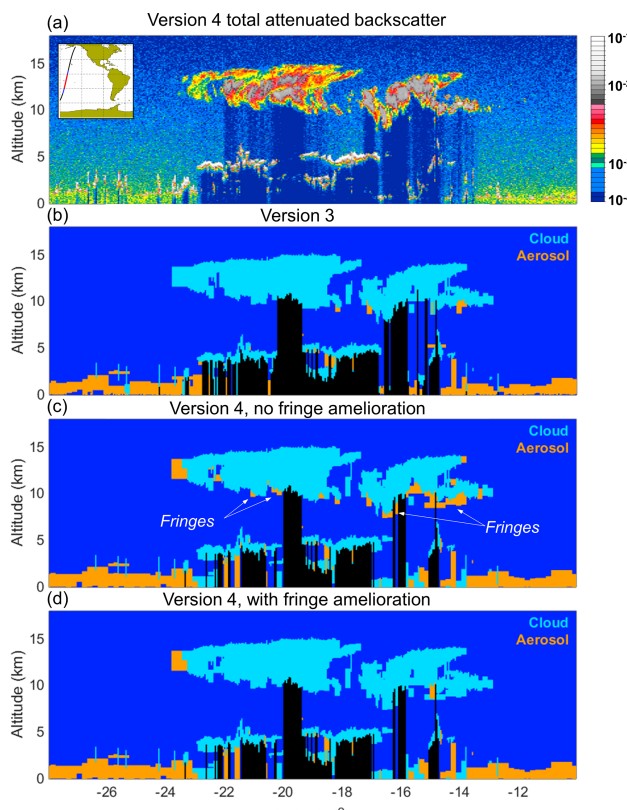

**Figure 6.** V4 532 nm total attenuated backscatter **(a)** and cloud–aerosol classification for V3 **(b)** and V4 without **(c)** and with **(d)** cirrus fringe amelioration for the granule 2008-06-01T12-27-28ZN. Red line on inset map shows approximate ground track.

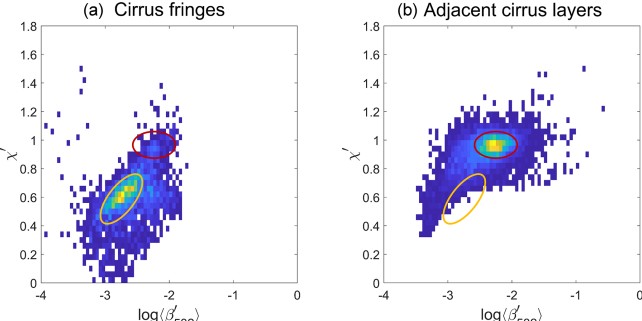

**Figure 7.** Layer-integrated attenuated color ratio, $\chi'$, vs. $\log_{10}$ of the 532 nm mean attenuated backscatter, $\log_{10}(\langle\beta'_{532}\rangle)$, for layers identified as cirrus fringes **(a)** and for adjacent cirrus layers **(b)**. Orange probability contours for cirrus fringes and red probability contours for the adjacent cirrus are overlaid in both panels **(a)** and **(b)**. Both plots show nighttime data acquired between 40° N and 40° S during January, February, and December 2008. The layer-integrated volume depolarization ratios for all layers, both fringes and cirrus, are in excess of 0.25.

Fig. 7 likewise exhibit distinctly lower $\chi'$ and $\log_{10}(\langle\beta'_{532}\rangle)$ values than the cirrus (this is evidenced by probability contours added in Fig. 7). In short, based solely on $\chi'$ and $\log_{10}(\langle\beta'_{532}\rangle)$, the optical properties of fringes are generally more similar to aerosols than clouds, and thus the possibility exists that at least some of these fringes are dust or smoke. However, by definition these fringes do not appear as discrete layers, but instead as areas that are in direct contact with a more strongly scattering layer that has been previously classified as an ice cloud. They are most often detected below semitransparent CE5 cirrus clouds, where the attenuation is large but the signal-to-noise ratio (SNR) remains larger than below attenuating cloud regions (see Fig. 6).

The fact that the color ratios in fringes are smaller than in the adjacent cirrus clouds cannot be explained by differential transmittance effects, as the same attenuation at both wavelengths is usually observed. One possible explanation for the change in optical properties seen in cirrus fringes is a reduction in size of the crystals linked to sublimation. The depolarization of the fringes is usually large (> 20 %), and fringes are quite frequently observed at latitudes far from dust sources, as demonstrated in Fig. 6. The pervasive presence of fringes and their immediate spatial proximity to layers previously identified as cirrus strongly suggest that these features are most likely also cirrus, and not aerosol layers.

To rectify these perceived misclassifications by the CAD algorithm, a "cirrus fringe amelioration" algorithm has been developed and added as a V4 CAD post-processor. To be identified as a cirrus fringe, a layer must (a) be initially classified by the CAD algorithm as an aerosol, (b) be detected at a 20 or 80 km horizontal averaging resolution, (c) be in direct contact with one or more layers detected at finer resolution and classified as cirrus, (d) have an attenuated backscatter centroid temperature below 0 °C, and (e) have a base altitude higher than 4 km above ground level. Layers that meet all of these criteria are classified as cirrus fringes and given a special CAD score of 106.

Application of these simple criteria shows that fringes are ubiquitous within the V4 dataset. For the data acquired between 60° N and 60° S during 2014, 22 % of all unique layers detected at 20 and 80 km averaging resolutions with bases between 4 and 16 km are identified as cirrus fringes. Note, however, that the fringe amelioration algorithm is not executed if, within an 80 km horizontal extent, 35 % or more of the features with bases above 4 km are originally classified as aerosol. This restriction prevents the amelioration algorithm from operating in those scenes that are likely to contain legitimate cases of clouds embedded in high-altitude aerosols.

Figure 6 demonstrates the routine impact of the fringe amelioration algorithm on CAD. These observations occurred over the remote South Pacific Ocean in June 2008, when dust and volcanic aerosols are not expected at high altitudes, as confirmed by back-trajectory analyses (not shown). Comparing the CAD in V3 (Fig. 6b) to that of V4 without cirrus fringe amelioration (Fig. 6c) shows an increase in the fraction of aerosols found adjacent to cirrus clouds.

After application of the cirrus fringe amelioration algorithm (Fig. 6d), the majority of the cirrus fringes are reclassified as cloud rather than aerosol. However, as demonstrated in Fig. 6d, some likely misclassifications of cirrus fringes remain in V4 due to limitations of the amelioration algorithm. Because their depolarization ratios are relatively large, these misclassified layers are most often classified as dust by the aerosol subtyping algorithm (Kim et al., 2018), which may introduce some bias in elevated dust occurrence (see further analyses in Sect. 5).

### 3.2.2 Corrections for water clouds lying beneath dense smoke

When applied to features detected at single-shot resolution, the V4 CAD algorithm can encounter difficulties in correctly assigning confidence levels to the classifications of dense water clouds lying beneath thick smoke layers. This situation often happens over the Atlantic Ocean off the west coast of southern Africa during the biomass burning season every year (June–September) (Remer et al., 2008; Chand et al., 2009; Das et al., 2017). The example shown in Fig. 8 occurred on 6 September 2008 at 1:35:29 UTC to the west of the African continent. Smoke attenuates the signal at 532 nm more strongly than at 1064 nm, leading to attenuated backscatter color ratios for the underlying water clouds that often far exceed the expected color ratio of $1.16 \pm 0.05$ that would be measured in the absence of overlying smoke (Vaughan et al., 2015). The spectrally dependent attenuation of the smoke is clearly evident from the attenuated backscatter color ratio measurements in Fig. 8b that increase dramatically with increasing penetration into the denser parts of the smoke layer (the color changes from orange to red to purple to gray between latitudes $\sim 6$ and $\sim 13° $ S).

Figure 9 shows the joint distribution of the overlying integrated attenuated backscatter and attenuated color ratio for the water cloud stratus deck below the extensive smoke plume from $\sim 18$ to $0.5° $ S. As can be seen, the attenuated color ratios can reach very high values (2 to 6 times the typical value) for these water clouds because of differential attenuation of the signal at the two CALIOP wavelengths. Because extinction coefficients are not retrieved for layers detected at CALIOP's single-shot resolution (Young and Vaughan, 2009), the attenuated backscatter coefficients in the water cloud cannot be corrected for the overlying signal attenuation from the smoke. Consequently, the water cloud color ratios are abnormally high and entirely inconsistent with the characterization of clouds with no overlying smoke layers. While these features are still classified as clouds, they are assigned very low CAD scores ($< 20$) and thus effectively transformed from high-confidence cloud layers identified in V3 (in which, by default, $\mathrm{CAD} = 100$ for all layers detected at single-shot resolution) into low- and no-confidence cloud layers in V4.

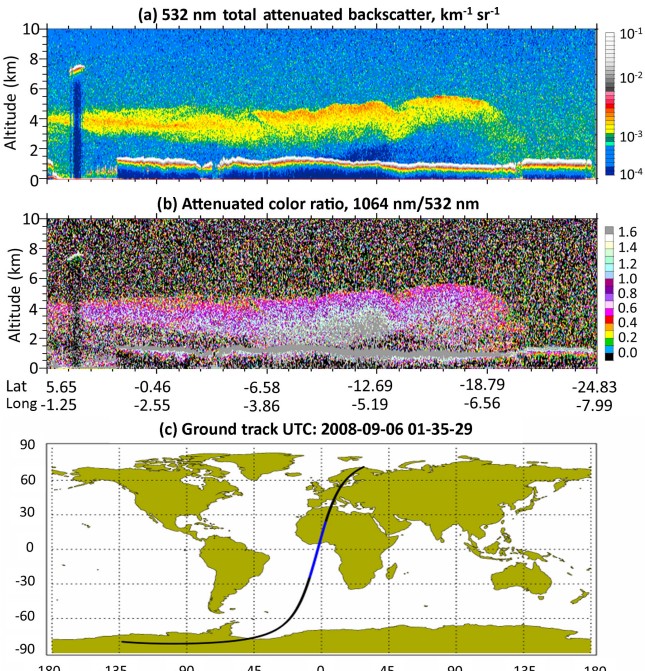

**Figure 8. (a)** Total attenuated backscatter and **(b)** 1064 nm / 532 nm attenuated color ratio and **(c)** the ground track on 6 September 2008 showing an extended smoke plume lying over a stratus cloud deck off the west coast of central Africa.

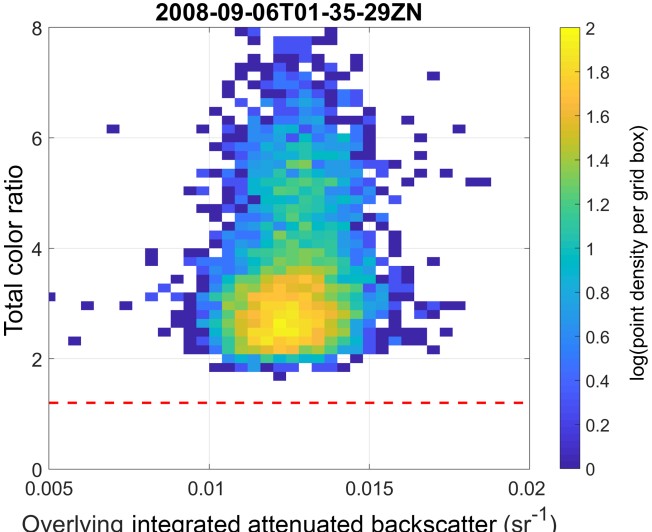

**Figure 9.** Joint distribution of integrated attenuated backscatter color ratio $\chi'$ of the water clouds and overlying 532 nm integrated attenuated backscatter $\gamma'$ above the water clouds for the 6 September 2008 data segment shown in Fig. 8. The dashed line corresponds to the value expected in the absence of overlying smoke aerosols.

The CAD algorithm is designed so that any layers that fall in the overlap region of aerosol and cloud PDFs or which have incorrect or unphysical parameters due to artifacts introduced in the measurement and/or data processing, as in

this case, are assigned low CAD values (Liu et al., 2009). However, the stratus deck over the South Atlantic has been widely observed and studied, and researchers worldwide are highly confident that these layers are unquestionably clouds (e.g., Sakaeda et al., 2011; Schrage and Fink, 2012). Because their identification as low- or no-confidence clouds by the V4 CAD algorithm would thus underestimate the true confidence in classifying these layers, a second CAD post-processor algorithm was designed to rectify the situation.

The classification errors arise primarily from the unphysical cloud color ratios that result from the differential signal attenuation at the two wavelengths as the laser pulses pass through the smoke layers. Therefore, following the assignment of initial CAD scores by the CAD algorithm, a special-purpose algorithm applies additional analyses to clouds with the following attributes: $0 \leq \text{CAD score} \leq 20$, $1.4 \leq \chi' \leq 10$, $\chi'$ relative uncertainty $< 500\%$, overlying layer-integrated attenuated backscatter at 532 nm ($\gamma'_{532}$) between 0.01 and 0.05 sr$^{-1}$, and mid-layer temperature $> 0\,^\circ$C. This set of parameters was chosen to select only those layers that might be classified as high-confidence water clouds were it not for their suspiciously high color ratios. For these layers, $\chi'$ is temporarily reset to 1.10 and the CAD score is recalculated. Only $\chi'$ is changed; all other parameters remain the same as in the original calculation. If the feature is still classified as a cloud in this second assessment by the CAD algorithm (a typical result, but not guaranteed), the CAD score is reset to the newly calculated value, and $\chi'$ is restored to its original value. Otherwise, the original CAD score remains in effect. Implementing this procedure effectively eliminated the majority of this kind of anomalous classification and was particularly effective in the smoke source region in southern Africa as well as the transport region off the coast.

### 3.2.3 Special CAD scores

As described in the opening paragraph of Sect. 3, nominal values for the CAD scores reported in the V4 CALIPSO data products range between −100 and 100. However, under special circumstances, the CALIOP scene classification algorithms will assign CAD scores that lie outside this range. These special CAD scores are enumerated in Table 1.

### 4 Overall comparison between V3 and V4 5 km layers

In this section, we present the overall changes in the CAD of various layers in V4 following the application of the new PDFs. Comparisons between V3 and V4 can be performed only for the tropospheric layers since in V3 the CAD algorithm was not applied to stratospheric layers or single-shot layers. Furthermore, the comparisons in this section are only made for layers detected at the 5 km averaging resolution because the PDFs were built based solely on these 5 km layers. Further bin-by-bin analyses based on the profile products,

which include all layers detected at the 5, 20, and 80 km horizontal averaging resolutions, are presented in Sect. 5. Layers detected at 20 and 80 km are generally tenuous features. Figure 10 shows the fractional occurrence of the CAD scores (panels a and b) and the V4 to V3 score ratios (panels c and d) for all the layers detected at 5 km horizontal resolution (cloud and aerosols) for the year 2008. The CAD scores for aerosols range from −100 to 0 and for the clouds from 0 to 100.

Table 2 compares the CAD scores for both data releases. For both aerosols and clouds during both day and night, the majority of the layers are being classified with a high degree of confidence (|CAD score| > 70) in both V3 and V4. The second column from the right-hand side of Table 2 shows that more than 90 % of all layers detected at 5 km are classified with a CAD score value greater than 70. However, the fraction of these highly confident classifications is ∼ 4 %– 5 % larger for V3 than V4. In Fig. 10, a large sharp spike is seen at CAD = 100 in the V3 cloud classifications for both day (∼ 69 %) and night (∼ 77 %). That is, more than two-thirds of all layers classified as cloud are assigned the highest possible confidence. As described earlier in Sect. 3, the V3 PDFs were built more conservatively for aerosols at higher altitudes because of the lack of measurement data. For the V3 PDFs shown in Fig. 5, the scaling factor $A_a$ in Eq. (2) was set to zero for aerosol layers with $\delta_v > 0.03$; that is, aerosols with $\delta_v > 0.03$ were assumed not to occur at high altitudes. As a consequence, the vast majority of high-altitude features (i.e., not only ice clouds, but also polarized volcanic aerosols, if they occurred at high altitudes, as well as artifacts and outliers) were classified as clouds with a CAD score of 100. This behavior contributes significantly to the sharp spike at CAD = 100 in the V3 distribution in Fig. 10.

The CALIOP June 2011 measurements shown in Fig. 4 demonstrate that volcanic ash can not only be lofted up to high altitudes, but can also exhibit large backscatter coefficients and depolarization ratios similar to those found in ice clouds. The construction of new V4 PDFs has specifically taken into account the occurrence of volcanic aerosols. The fraction of the clouds with CAD = 100 has decreased significantly (to ∼ 20 %) in V4 compared with V3 (Fig. 10c and d). The two most common CAD scores in V4 occur at CAD = −99 and CAD = 99. The total fraction for a CAD range of 98 to 100 is 59.1 % and 66.9 % in V4 for night and day, respectively, which are ∼ 15 % smaller than the corresponding values of 74.0 % and 83.8 % in V3. It appears that the cloud CAD score may have been generally overestimated in V3, especially at high altitudes. The bumps in Fig. 10 between CAD scores of 0 and ∼ 20 correspond to outliers that either have highly suspicious layer optical properties or represent unusually large noise excursions or other artifacts.

**Table 1.** Special CAD scores that can be reported in the CALIPSO V4 lidar level 2 data products. The occurrence frequencies are obtained from an analysis of all unique layers detected at 5, 20, and 80 km averaging resolutions from 2013 to 2015 ($N = 129\,098\,950$). n/a: not applicable.

| Value | Occurrence frequency | Interpretation |
|---|---|---|
| $-101$ | $6.2 \times 10^{-8}$ | When evaluating a layer detected at the 5 km averaging resolution, the scene classification module encountered a negative value for the layer 532 nm mean attenuated backscatter, $\langle \beta'_{532} \rangle$. These layers should be considered artifacts and excluded from all science analyses. |
| 101 | n/a | Used in version 2 data products only; obsolete in later versions |
| 102 | n/a | Used in version 2 data products only; obsolete in later versions |
| 103 | $\sim 0.010$ (i.e., $\sim 1\,\%$ of all layers detected) | The layer-integrated attenuated backscatter at 532 nm ($\gamma'_{532}$) is suspiciously large. While the spatial properties and volume depolarization ratios of these layers are generally reliable, all other optical properties should be excluded from scientific studies. The most likely cause of these very large values of $\gamma'_{532}$ is an overestimate of the optical depths of overlying layers (Young and Vaughan, 2009). |
| 104 | $1.4 \times 10^{-4}$ | These are the lowest layers detected in profiles averaged to 5 km horizontal resolution. These layers are classified as opaque at the 5 km resolution. However, there are profiles within the 5 km average in which no layer was subsequently detected at single-shot resolution, indicating that total signal attenuation at single-shot resolution occurs at some higher altitude. The spatial properties of these layers are highly reliable, but their optical properties are less so. |
| 105 | $2.1 \times 10^{-5}$ | A negative value of $\langle \beta'_{532} \rangle$ was encountered in the scene classification module for a layer detected at an averaging resolution other than 5 km. Layers with CAD = 105 that are detected at the single-shot and 1 km resolutions should be considered artifacts and excluded entirely from all science analyses. For layers detected at the 20 and 80 km averaging resolutions, negative values are most likely introduced by the attenuation corrections applied to account for the optical depths of overlying layers. The spatial properties of these layers are generally reliable, but their optical properties should be excluded from scientific studies. |
| 106 | $\sim 0.015$ (i.e., $\sim 1.5\,\%$ of all layers detected) | The layer was originally classified as aerosol but later reclassified as ice cloud by the fringe amelioration algorithm described in Sect. 3.2.1. Both the spatial and optical properties of these layers are generally reliable and suitable for inclusion in scientific analyses. |

## 5 Assessment of V4 CAD performance

The analyses in the earlier sections were applied only to layers detected at a 5 km horizontal resolution because only those 5 km layers were used in constructing the V4 PDFs. In this section, we primarily use the 5 km profile products, which allow us to assess CAD performance on a range bin-by-range bin basis. These more comprehensive analyses include all features detected at 5, 20, and 80 km resolutions and thus provide greater insight into the V4 CAD performance. Note, however, that these analyses are confined exclusively to assessments of CAD and do not explore the performance of the aerosol subtyping algorithms or the cloud ice–water phase determination scheme. Those investigations are instead described in separate publications: Kim et al. (2018) for aerosol subtyping and Avery et al. (2018)

for ice–water phase. Additionally, an independent assessment of CALIOP CAD performance is conducted by Zeng et al. (2018), who use an unsupervised machine-learning technique (i.e., a fuzzy $k$-means clustering algorithm) to distinguish clouds from aerosols and then compare and contrast their results to the classifications and CAD scores reported in the CALIOP V4 data products.

### 5.1 V4 CAD in the troposphere

#### 5.1.1 Case studies

**Dense dust layers over the Taklimakan Desert**

The Taklimakan, located in the Tarim Basin in northwest China at about $40°$ N, is one of the world's major deserts and most prolific dust sources (Prospero et al., 2002). The

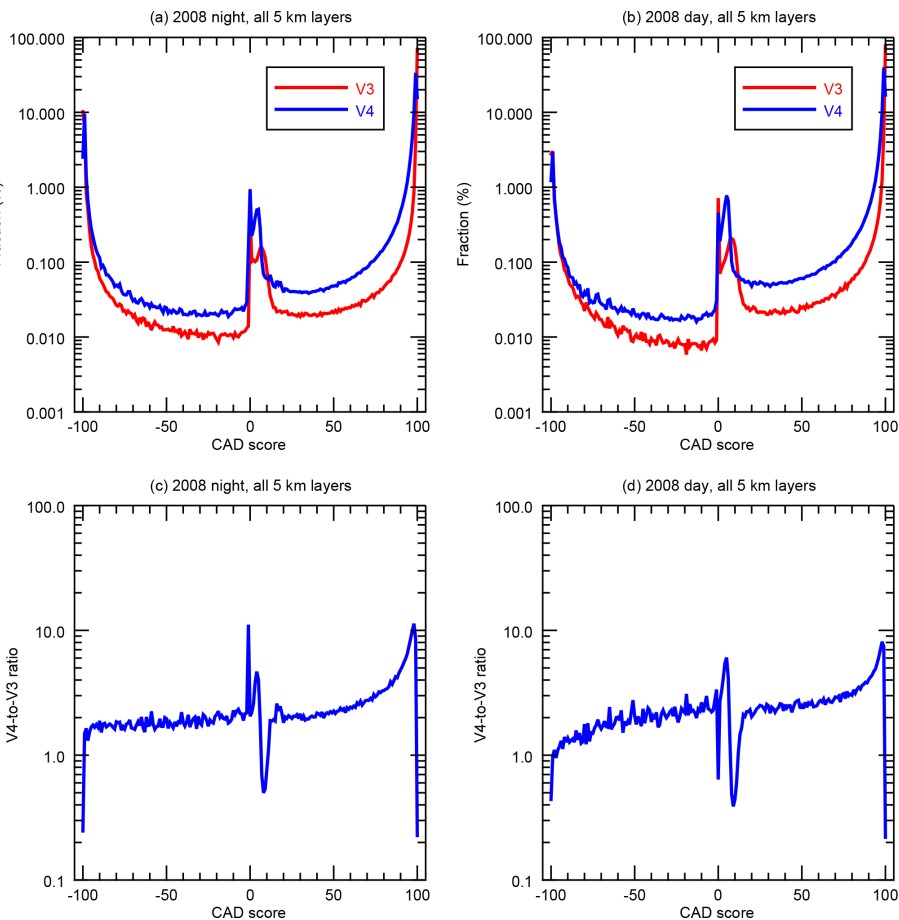

**Figure 10.** Distribution of occurrence frequencies **(a, b)** and their ratios **(c, d)** as a function of CAD score for all tropospheric layers detected at a 5 km resolution for 2008 V3 and V4, for nighttime **(a, c)** and daytime data **(b, d)**.

**Table 2.** Comparison of V3 and V4 CAD scores for all 5 km layers for the year 2008.

|  | Aerosol faction (%) | | | | Cloud fraction (%) | | | | High confidence (%) | Total layers |
|---|---|---|---|---|---|---|---|---|---|---|
| CAD | $-100$ | $-98$ to $-100$ | $< -70$ | All | 100 | 98 to 100 | $> 70$ | All | $< -70$ or $> 70$ | |
| V3 night | 10.2 | 17.5 | 19.3 | 20.3 | 69.3 | 74.0 | 76.5 | 79.7 | 95.8 | $1.815 \times 10^7$ |
| V4 night | 2.5 | 13.5 | 16.4 | 18.4 | 15.7 | 59.1 | 74.6 | 81.6 | 91.0 | $1.770 \times 10^7$ |
| V3 day | 2.8 | 6.5 | 8.2 | 9.0 | 77.1 | 83.8 | 87.1 | 91.0 | 95.3 | $1.713 \times 10^7$ |
| V4 day | 1.2 | 4.9 | 7.0 | 8.6 | 17.0 | 66.9 | 83.2 | 91.4 | 90.2 | $1.689 \times 10^7$ |

dust activity over the Tarim Basin area is persistent almost all year long, reaching a maximum in the spring (Z. Liu et al., 2008b; D. Liu et al., 2008). The Tarim Basin is surrounded by high mountains, with the Tian Shan mountains in the north and the Kunlun Mountains in the south and southwest. These mountains create circulations in the basin that are favorable for dust to remain suspended aloft for long periods of time (Tsunematsu et al., 2005). Taklimakan dust can often reach altitudes high into the troposphere and subsequently be transported long distances by westerlies (Huang et al., 2008; Uno et al., 2009).

One of the issues with the V3 CAD was that dense dust layers over the Taklimakan area were often misclassified as cloud when they were lofted to relatively high altitudes and/or transported far to the north ($> \sim 40°$ N), where the occurrence of ice clouds becomes more significant compared to dust (Jin et al., 2014). In these cases, classification skill has been improved in the V4 CAD algorithm by reducing the latitude bands of the PDFs from 10° in V3 to 5° and optimizing the height-dependent characteristic scattering parameters used in the PDFs.

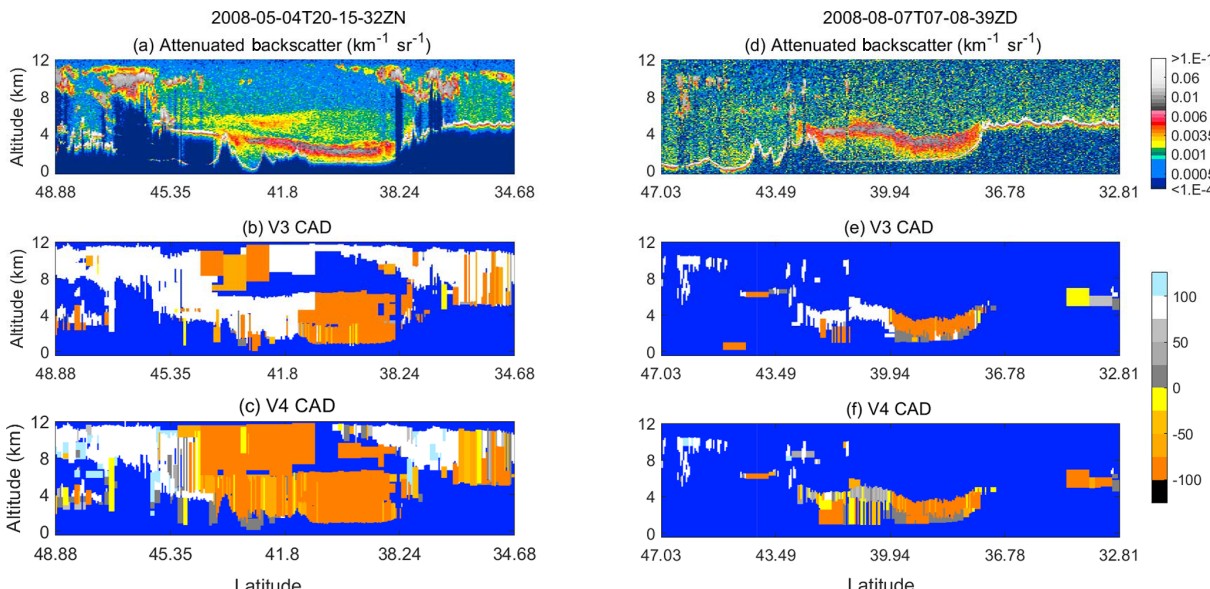

**Figure 11.** The 532 nm attenuated backscatter coefficients **(a, d)**, V3 CAD scores **(b, e)**, and V4 CAD scores **(c, f)** for the granule 2008-05-04T20-15-32ZN located between 31 and 49° N **(a–c)** and for the granule 2008-08-07T07-08-39ZD located between 29 and 47° N **(d–f)**.

Dense dust layers detected at single-shot resolution were classified as cloud by default in V3. In V4, the CAD algorithm is now also applied to single-shot layers. This extended application of the CAD algorithm has significantly reduced the misclassification of single-shot dust layers. An example of dense dust over Taklimakan, observed by CALIOP around 20:15:32 UTC on 4 May 2008, is shown in Fig. 11a–c. Multilayered dust (yellow–red–grayish areas) appears between 38.24 and 45.4° N and extends from the surface to ∼ 12 km, with the densest layer (red–grayish area) at 2–4 km. An attenuating water cloud is embedded in the dust at ∼ 4 km north of 43.6° N. While the V3 CAD algorithm correctly identified the water cloud, a large portion of the dust north of ∼ 40° N was misclassified as high-confidence cloud (mostly as ice cloud). As can be seen in Fig. 11c, the V4 CAD correctly classifies these layers as aerosols with good confidence levels (CAD scores between −100 and −50).

Another example of dense dust over Taklimakan, which occurred at 07:08:39 UTC on 7 August 2008, is shown in Fig. 11d–f. In this example, part of the heavy dust portion at ∼ 4–5 km is misclassified as cloud in V3 and remains misclassified as cloud in V4, but with much lower CAD scores. Very dense dust layers located above ∼ 4 km and north of ∼ 40° N will sometimes be misclassified as cloud because north of ∼ 40° N a significant amount of ice cloud exists at altitudes of 4 km and below. The V3 misclassification of high-altitude and high-latitude dense dust is not completely corrected in V4, as Fig. 11 illustrates, but the frequency of these dense dust cases is very low.

When a dust layer is misclassified as cloud, the high layer depolarization ratio can cause it to be further misclassified as an ice cloud by the cloud-phase algorithm (Hu et al.,

2009; Avery et al., 2018), and hence the CALIPSO data products can sometimes report ice clouds that have temperatures warmer than 0 °C (Liu et al., 2009). These so-called "hot cirrus" (e.g., Liu et al., 2009) are highly likely to be misclassifications of dust. To more quantitatively evaluate the changes of the V4 CAD compared to the V3 CAD, Fig. 12 shows seasonal variations in (a) V3 aerosol fraction, (b) V4 aerosol fraction, (c) V3 clouds that changed to V4 aerosol, (d) V3 hot cirrus fraction, (e) V4 hot cirrus fraction, and (f) V3 hot cirrus clouds that changed to V4 aerosol for a selected geographic region (35 to 45° N, 75 to 90° E) that contains the entire Tarim Basin where dust is the dominant aerosol type (Wang et al., 2008). Hot cirrus occurs most frequently between 3 and 5 km, where dense dust can be lofted and clouds start to occur more frequently. The column mean of V3 hot cirrus fraction reaches a maximum of ∼ 8 % in August 2008 (Fig. 12d), and close to 80 % of the V3 hot cirrus changed to V4 aerosol (Fig. 12d and f). The hot cirrus fraction in V4 is significantly reduced, reaching a monthly maximum value of less than 3 % during the summer (Fig. 12e). This indicates that V4 has been improved significantly in this particular geographic region, especially at those altitudes where dense dust is most frequently misclassified as cloud in V3. There should also be a certain fraction of misclassified dust with temperatures colder than 0 °C that was classified as ice cloud by the cloud-phase algorithm in V3 and still remains misclassified as ice in V4. However, it is very difficult to quantify this type of misclassified dust.

Overall, there are more aerosols in V4 than in V3 because many of the layers misclassified as clouds in V3 are correctly classified as aerosols in V4. These changes occur mainly at relatively high altitudes, as seen in Fig. 12c, and reach a max-

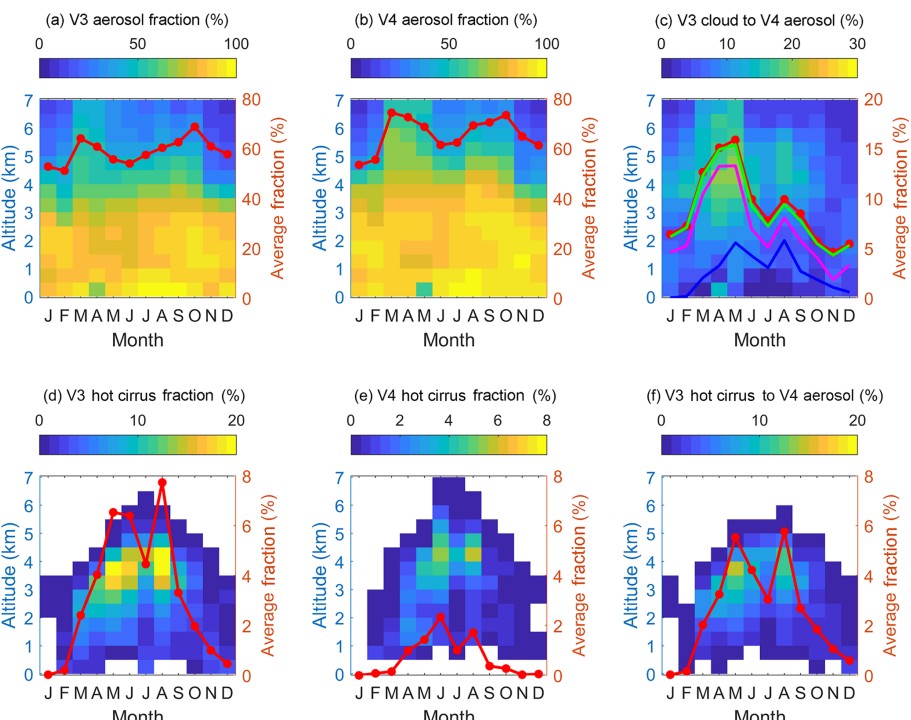

**Figure 12.** Seasonal variations in **(a)** V3 aerosol fraction, **(b)** V4 aerosol fraction, **(c)** fractions of V3 clouds changing to V4 aerosols, **(d)** V3 hot cirrus, **(e)** V4 hot cirrus, and **(f)** the fraction of V3 hot cirrus that changes to V4 aerosol in the Taklimakan region (35 to 45° N, 75 to 90° E). The fractions in panels **(a)**, **(c)**, **(d)**, and **(f)** are relative to the total number of V3 clouds and aerosols whereas the fractions in panels **(b)** and **(e)** are relative to the V4 total. The red curve in each panel is a column average fraction (%) between 0 and 7 km. Additional information is provided in panel **(c)**, in which the blue curve shows the contributions from V3 hot cirrus and the magenta curve shows the contributions from all V3 ice clouds. The difference between the green and magenta curves quantifies the contributions of no-confidence clouds, while the difference between the red and green curves quantifies the (very small) contributions from V3 water clouds. In total, 29.6 % of all V3 cloud-to-V4 aerosol changes were considered hot cirrus in V3. High-confidence ice clouds, low-confidence ice clouds, and water clouds contribute 47.1 %, 19.2 %, and 4.1 %, respectively.

imum of ∼ 20 % at altitudes of 4–5 km during the spring. For the column average from 0 to 7 km over this geographic region, the fractional change from cloud to aerosol varies from 5 % in the late fall to 16 % in the summer. The change from V3 ice contributes the most and accounts for 47.1 % of the total V3 cloud to V4 aerosol change, followed by the change from V3 hot ice CE6 (29.6 %) and V3 no-confidence cloud (19.2 %). Only a very small fraction of V3 water clouds changed their type, accounting for 4.1 % of the total change.

**Lofted Asian dust layers near the Arctic**

The V4 CAD algorithm also makes significant improvements in classifying lofted layers of Asian dust and polluted dust that are transported to the Arctic each spring. The Arctic regions have long been known to be impacted by aerosols from midlatitude sources, with the primary evidence being the springtime haze in the lower troposphere (Garrett and Verzella, 2008). Pollution and dust from Asian sources can reach the Arctic in 3 to 5 days, carried by midlatitude cyclones (Di Pierro et al., 2011, 2013; Z. Huang et al., 2015). In the earlier CALIOP data product releases, dust layers over

the Arctic could be misclassified by the CAD algorithm as ice clouds (Di Pierro et al., 2011). An example measured at 18:28:54 UTC on 1 March 2008 is shown in Fig. 13. The attenuated backscatter data in Fig. 13a show numerous faint layers visible at altitudes of 5 to 10 km between 50 and 80° N. These layers show enhanced depolarization, albeit with depolarization ratios smaller than the typical values for ice clouds. Back trajectories analyzed using the HYSPLIT model from a representative point (74° N, 138° E) along the CALIPSO transect (Fig. 13d) indicate that most of these air parcels were originated and lifted up from the surface close to the Taklimakan Desert within the previous 5 days. These layers are thus likely dust transported from the lower latitudes, as demonstrated by Di Pierro et al. (2011, 2013). Many of these layers were misclassified as ice clouds in V3, as shown in Fig. 13b, and are now correctly classified in V4 as aerosols with high CAD scores, as shown in Fig. 13c.

In polar winter, ice crystals often form in clear skies when temperatures become very cold, due to slow isobaric cooling of moist air advected from lower latitudes, especially over the Antarctic plateau (Lachlan-Cope, 2010). Crystal concen-

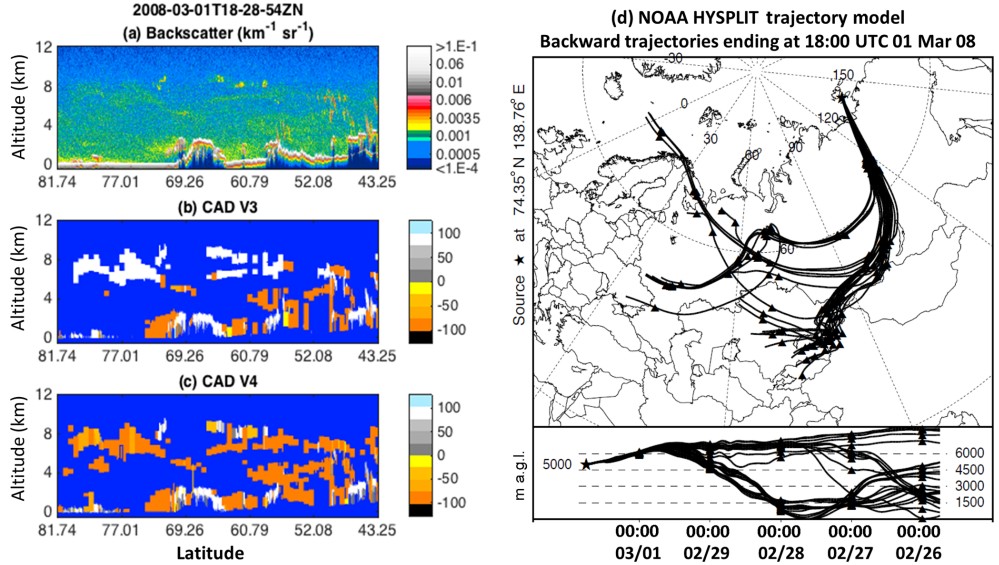

**Figure 13.** The (a) 532 nm attenuated backscatter coefficients, (b) V3 CAD scores, and (c) V4 CAD scores for the granule 2008-03-01T18-28-54ZN between 43 and 81° N, and (d) NOAA HYSPLIT back trajectories starting at 74.35° N, 138.76° E.

trations tend to be low and were often misclassified as aerosol in V3, although in V3 aerosol in polar regions above ice and snow surfaces could only be classified as clean or polluted continental. In V4, the misclassification of tenuous ice crystals as aerosol still occurs, as pointed out by di Biagio et al. (2018). Note that in V4, the restrictions on aerosol type in polar regions have been removed and these tenuous ice crystals tend to be classified in V4 as mineral dust when misclassifications occur, even in regions where detectable intrusions of mineral dust from midlatitude source regions are rare.

### High-altitude smoke – the Black Saturday event

Smoke plumes are often found in the upper troposphere and are occasionally injected above the tropopause by pyrocumulonimbus convection triggered by fires (Fromm et al., 2010; de Laat et al., 2012; Khaykin et al., 2018). A prominent example is the Black Saturday plume from the bushfires of Australia on 7 February 2009 (de Laat et al., 2012; Glatthor et al., 2013). Figure 14 shows smoke plumes at high altitudes $\sim 20$ to $\sim 40°$ S on 10 February 2009. The low depolarization ratio (below 6 %, Fig. 14b) and increasing color ratio from top to base (not shown here) suggest that these layers are smoke, in contrast to the cloud layers with large depolarization and relatively uniform color ratio between $\sim 2$ and $\sim 20°$ S. Whereas most of these layers were classified as clouds by the V3 CAD, the V4 CAD now correctly identifies these high-altitude smoke layers as aerosols. Note that the plume at lower altitudes between $\sim 20$ and $\sim 28°$ S has high confidence (large CAD scores) in V4 and the plume at high altitudes between $\sim 30$ and $\sim 40°$ S has low or no confidence (small CAD values). This is an indication that the probability

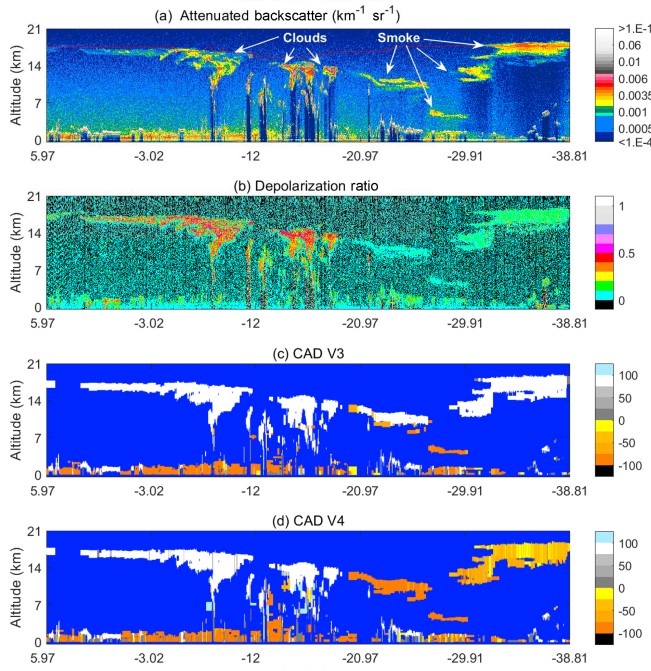

**Figure 14.** (a) V4 532 nm attenuated backscatter coefficients, (b) 532 nm volume depolarization ratios, (c) version 3 CAD scores, and (d) version 4 CAD scores for the granule 2009-02-10T12-33-03ZN between 5.97° N and 38.81° S.

for aerosols to be present at higher altitudes is smaller than at lower altitudes.

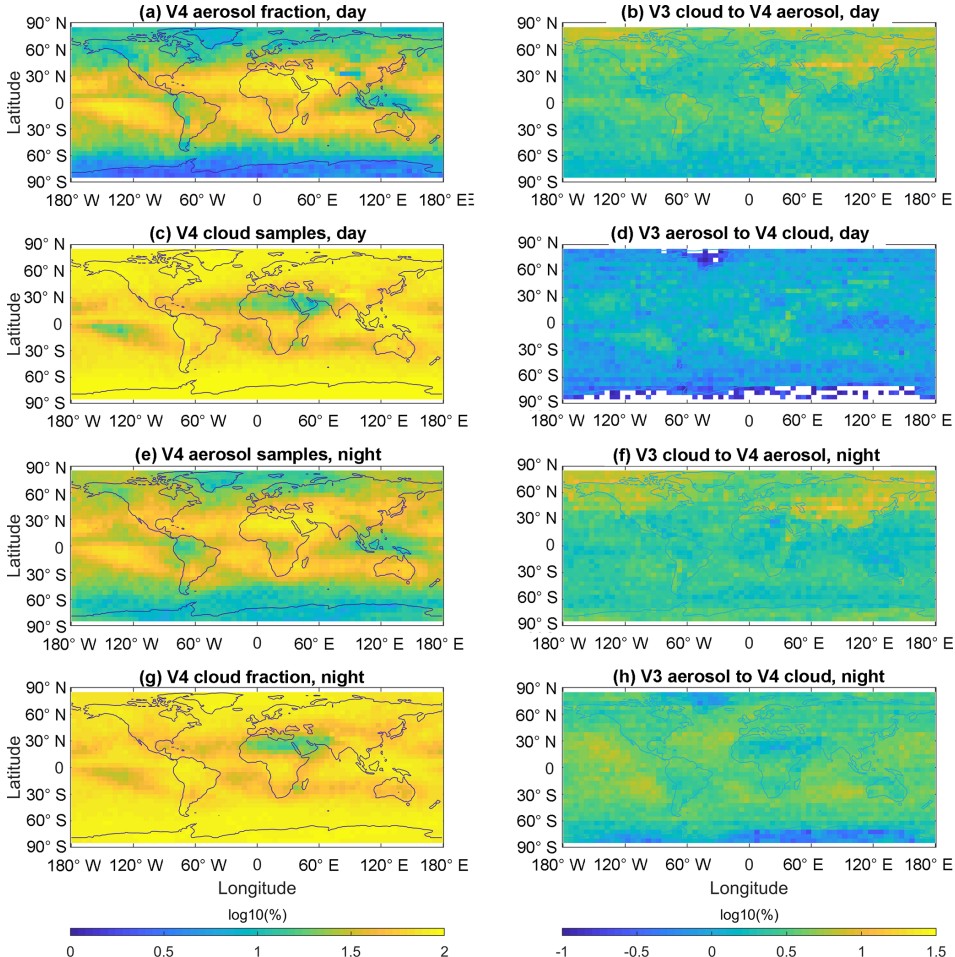

**Figure 15.** Geographical distributions of V4 aerosol and cloud fractions (in $\log_{10}$ percentage) derived from 1 year (2008) of the CALIOP day **(a–d)** and night **(e–h)** profile products. Percentages are computed by dividing the number of $60\,\text{m} \times 5\,\text{km}$ range bins classified as aerosol **(a, e)** or cloud **(c, g)** by the total number of range bins containing either aerosol or cloud. The corresponding fractional changes from V3 cloud to V4 aerosol are shown in panels **(b)** and **(f)**. Panels **(d)** and **(h)** show the fraction changes from V3 aerosol to V4 cloud.

**Table 3.** Scene classification confusion matrices (V3 vs. V4) for the year 2008, **(a)** nighttime and **(b)** daytime. The third column of the first two rows is the fraction of the V3 cloud or aerosol relative to the total number of V3 features detected. The first two columns of the third row show the percentage of the V4 cloud or aerosol relative to the total number of V4 features detected.

| **(a)** Night | Cloud | Aerosol | V3 total |
|---|---|---|---|
| Cloud | 95.4 | 4.6 | 70.8 |
| Aerosol | 11.1 | 88.9 | 29.2 |
| V4 total | 70.8 | 29.2 | 93.5 |
| **(b)** Day | Cloud | Aerosol | V3 total |
| Cloud | 95.7 | 4.3 | 76.3 |
| Aerosol | 4.6 | 95.4 | 23.7 |
| V4 total | 74.1 | 25.9 | 95.6 |

## 5.1.2 Global statistics

### Confusion matrices

Table 3 shows scene classification confusion matrices for the year of 2008, calculated from individual range bins within the troposphere obtained from the 5 km profile products. These comparisons use only those range bins that are classified as either cloud or aerosol in both V3 and V4. Range bins reporting new features detected in V4 only and range bins that report features detected in V3 that were not detected in V4 are excluded. The first two rows represent the V3 clouds and aerosols and the first two columns indicate the V4 clouds and aerosols. The two diagonal elements represent the percentage of range bins that remain unchanged; the off-diagonal elements show the percentage of range bins for which the feature type changed. The third column of the first two rows is the percentage of the V3 cloud or aerosol rel-

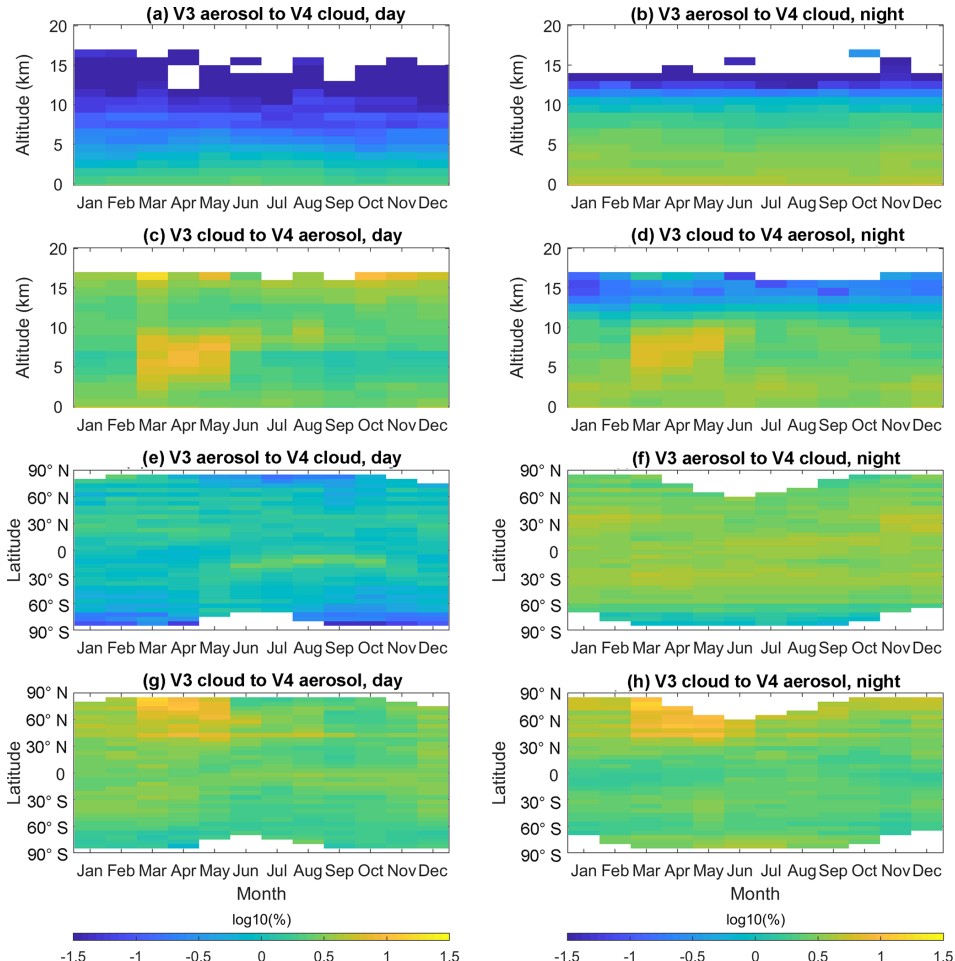

**Figure 16.** Seasonal distributions of fractional changes of V3 aerosol to V4 cloud or V3 cloud to V4 aerosol as a function of altitude **(a–d)** and latitude **(e–h)** (**c**, day; **d**, night), and seasonal variations in latitudinal changes of version 3 aerosol switching to version 4 cloud (**e**, day; **f**, night), version 3 cloud switching to version 4 aerosol (**g**, day; **h**, night).

ative to the total number of V3 features detected. Similarly, the first two columns of the third row give the percentages of the V4 cloud or aerosol relative to the total number of V4 features detected. The diagonal element of the third row and third column is the percentage of the total number of range bins that remains unchanged.

We see from this table that, for both daytime and nighttime, less than 5 % of V3 clouds are changed to aerosols in V4. While 4.6 % of V3 daytime aerosols are reclassified as clouds in V4, this change is more than 2 times larger (11.1 %) at night. Our preliminary analyses suggest that more cirrus fringes are detected at night than during the day, presumably because of the better nighttime SNR. Overall, the net cloud and aerosol fractions remained the same during the night, whereas the V4 net aerosol faction increased by $\sim 2$ % during day.

Because the lower 532 nm calibration coefficients in V4 increase the magnitude of the attenuated backscatter coefficients, the V4 feature finder detection totals increased by

12.6 % and 6.2 %, respectively, at night and during the day (refer to Table A1 in Appendix A). At the same time, about 3.6 % of cloud and aerosol features reported in V3 during both night and day were not detected in V4, resulting in a net increase of 9.0 % in the V4 nighttime data and 2.5 % in the daytime data. The new features that are excluded in Table 3 are classified about equally as clouds and aerosols in V4.

## Geographical distributions and seasonal variations

To investigate possible spatial patterns in the CAD classification changes, we used the same 5 km profile products to locate where and when the classification changes occurred. Figure 15 presents geographic distributions of V4 cloud and aerosol fractions and the corresponding fractional change of V3 aerosol to V4 cloud or V3 cloud to V4 aerosol relative to the total V3 cloud and aerosol. The distribution patterns of the changes essentially follow the patterns of the cloud and aerosol distributions. More changes of V3 cloud to V4

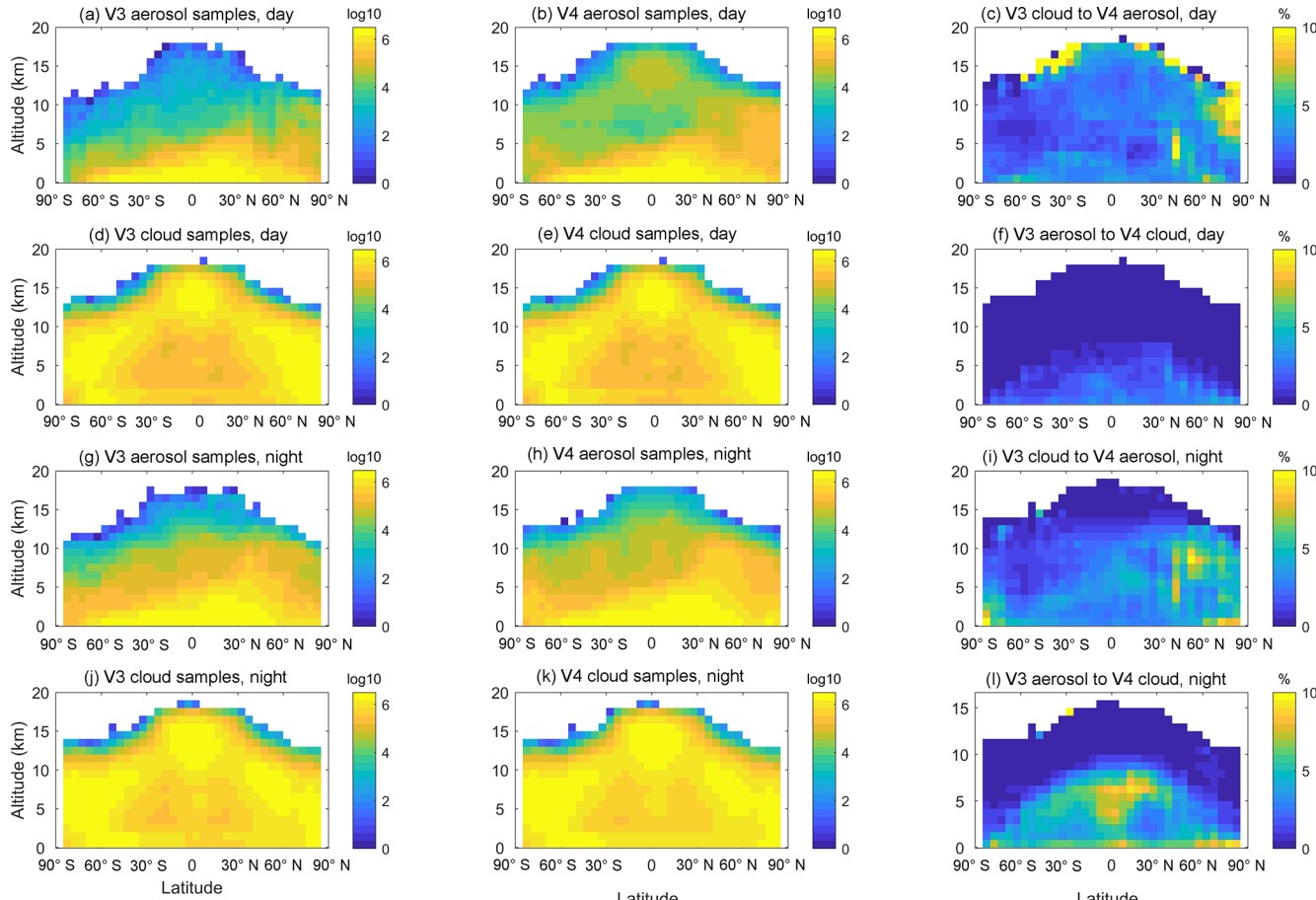

**Figure 17.** Altitude–latitude distributions of aerosol and cloud scene samples of V3 **(a, d, g, j)** and V4 **(b, e, h, k)** as well as the fractional changes **(c, f, i, l)** of the total V3 cloud and aerosol scenes derived from 1 year (2008) of the CALIOP day **(a–f)** and night **(g–l)** profile products.

aerosol are seen in the dust and smoke regions (Fig. 15b and f). As shown in Fig. 16a and b, the fractional change of V3 aerosol to V4 cloud decreases with increasing altitude because the aerosol occurrence is relatively low at higher altitudes. However, the fractional change relative to the V3 aerosol is very large at high altitudes (not shown). The significant changes from V3 clouds to V4 aerosols that are seen at 5–10 km appear to correspond to Asian dust activity over the sources and transport to the Arctic (during March–May; see Fig. 16c and d) or smoke plumes in the central and southern Africa (during August–October; see Fig. 16g and h).

Figure 17 presents joint altitude–latitude distributions of V3 and V4 aerosols and clouds in the left and middle columns, respectively. In Fig. 17b there appears to be a mode in the V4 daytime aerosol distribution in the tropical upper troposphere that shows a correlation to the tropical cloud distribution in Fig. 17e. This is mainly a residual of ice fringe candidates that were not changed to ice after applying the fringe amelioration algorithm. Shown in the right column in Fig. 17 are the changes of V3 aerosol to V4 cloud or V3

cloud to V4 aerosol relative to the total number of clouds and aerosols in each grid. Although most of the V3 aerosols at altitudes above ∼ 10 km have been changed to clouds in V4, more high-altitude V3 clouds were converted to aerosols in V4. In general, this behavior is expected, as the V4 CAD PDFs were deliberately designed to be more sensitive to the presence of lofted high-altitude aerosols. However, when misclassifications occur, the residuals are most often classified as dust by the aerosol subtyping algorithm (Omar et al., 2009; Kim et al., 2018).

There does not appear to be a clear high-altitude tropical aerosol mode in the nighttime V4 aerosol distribution in Fig. 17h, providing evidence that the mode seen in the daytime data is the result of classification errors. Our analysis shows that there are about 3 times more high-altitude layers detected at 5 km resolution and subsequently classified as dust during the daytime than at night. This can partly explain the day and night difference in the V4 aerosol distributions seen in Fig. 17 because layers detected at 5 km resolution are not processed by the fringe amelioration algorithm.

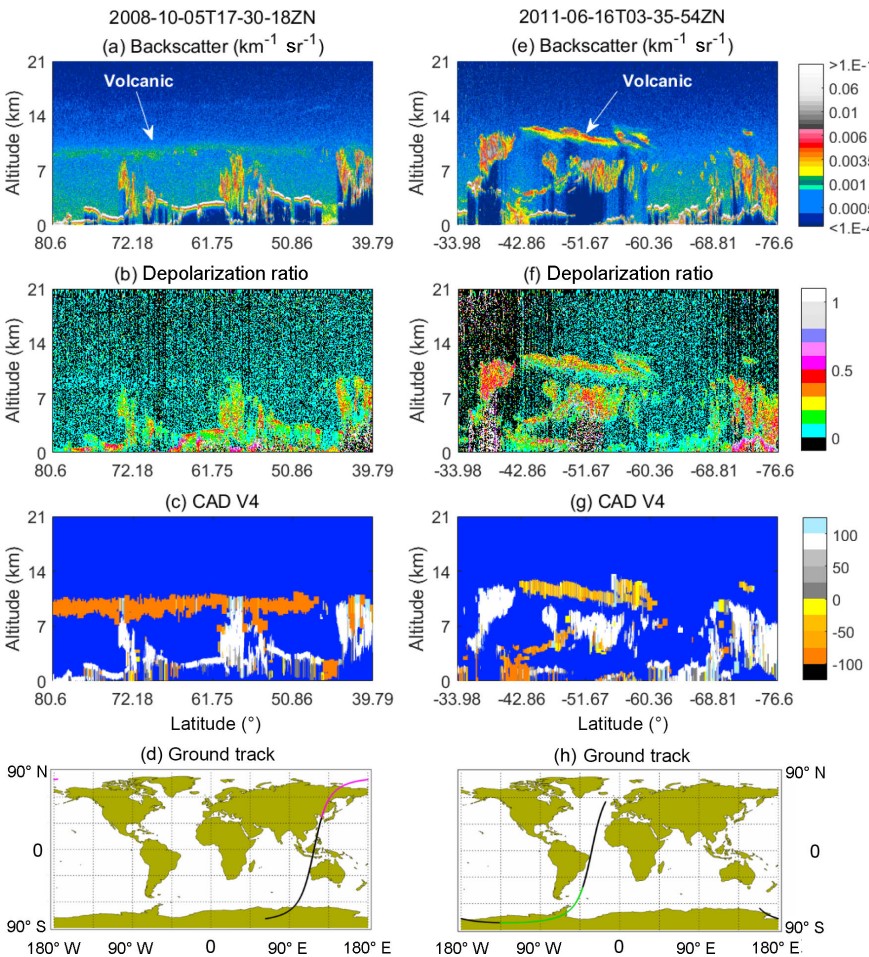

**Figure 18.** Examples of volcanic layers in CALIPSO data. V4 532 nm attenuated backscatter **(a, e)**, 532 nm volume depolarization ratios **(b, f)**, and CAD scores **(c, g)** of stratospheric volcanic layers and ground track **(d, h)** from the June 2011 Puyehue-Cordón Caulle **(a, b, c, d)** and August 2008 Kasatochi eruptions **(e, f, g, h)**.

## 5.2 CAD performance in the stratosphere

### 5.2.1 Stratospheric volcanic aerosol

The CAD algorithm was originally designed for cloud and aerosol layers in the troposphere and in previous versions did not include data from the higher altitudes in its training set. As such, layers detected in the stratosphere were not characterized in versions prior to V4 and were instead simply called stratospheric features. The training set used to develop the V4 CAD PDFs included stratospheric data from all of 2008 and from two volcanic eruptions in June 2011 (see Sect. 3). These PDFs were subsequently applied to all layers detected in the stratosphere in the V4 operational algorithm, thus enabling the classification of volcanic and other stratospheric aerosol and cloud layers. Figure 18 shows two examples of V4 CAD performance when classifying stratospheric volcanic layers.

The extensive and clearly visible layer of enhanced backscatter between ∼ 50 and ∼ 80° N in Fig. 18a is a vol-

canic layer injected by the Kasatochi eruption (55° N) in August 2008 (Vernier et al., 2013). The persistence of this layer during the 3 months after the eruption underscores the need to characterize these layers properly. $\delta_v$ for this extended layer was relatively low, suggesting the predominance of sulfate particles (Fig. 18b). As can be seen in Fig. 18c, the V4 CAD algorithm correctly classified almost all of the volcanic layer as aerosol with high confidence (CAD scores approaching ∼ −100). Figure 18e shows another plume resulting from the eruption of Puyehue-Cordón Caulle in Chile in June 2011 at an altitude of 10–12 km. In contrast to the Kasatochi volcano, the silicate ash content in this plume was very high (Vernier et al., 2013), as can be seen in the relatively high $\delta_v$ in Fig. 18f. Once again, the V4 CAD algorithm classifies most of the layers as aerosols. However, the CAD scores are not very high (Fig. 18g). This largely reflects the fact that the probability for a relatively dense depolarizing aerosol to be present at high altitudes is low, and the cloud and aerosol PDF overlap region here is large compared with low alti-

tudes. A significant fraction of the Puyehue-Cordón plume is classified as cloud, possibly due to high color ratios and depolarization ratios that fall within the PDF overlap region with ice clouds. Because volcanic eruptions can release large amounts of water vapor, which can then condense into ice cloud particles (Guo et al., 2004), it is not always possible to determine with absolute certainty whether the putative cloud layers in the Cordón plume are misclassified aerosols or actually legitimate clouds.

While the V4 CAD can distinguish aerosols and clouds for stratospheric layers, uncertainties tend to increase as the altitude increases. This increasing uncertainty derives from the fact that the very low aerosol occurrence frequency at high altitudes does not provide a statistically significant sample size to constrain the PDFs, and thus the high-altitude PDFs were created by extrapolation from measurements at lower altitudes. Further, because the SNR of stratospheric layers is typically quite low, there is a widening in the distribution of color ratio and attenuated backscatter for stratospheric features compared to features at lower altitudes, leading to generally lower stratospheric CAD scores. This can be seen in Fig. 19, which shows the CAD scores of both aerosol and cloud layers with bases within 2 km above the tropopause and above 4 km above the tropopause for July–October 2008 between $\sim 50$ and $\sim 82°$ N. These layers are mostly from the Kasatochi volcano. Within 2 km above the tropopause, the CAD algorithm classifies both aerosols and clouds with good confidence. However, as we go higher up in the stratosphere, the general lack of data, as well as decreasing SNR for weaker features, makes CAD increasingly difficult. Furthermore, the fraction of feature-finder false positives may become significant, especially for daytime measurements over bright surfaces or optically thick stratus cloud decks. These false positives generally have very small CAD scores, which quite rightly reflect a lack of classification confidence. As a result, at very high altitudes, most of the layers classified as clouds exhibit very low or no confidence (CAD score < 20) (similar to aerosols, as seen in Fig. 22) and the CAD algorithm generally seems to provide somewhat more confidence in the aerosol classification than the clouds. This is consistent with the general dearth of cloud occurrence at stratospheric altitudes.

### 5.2.2 Polar stratospheric clouds and aerosols

PSCs are ubiquitous in both polar regions in local winter and have important consequences for polar ozone loss processes (e.g., Lowe and MacKenzie, 2008). Along with other stratospheric layers, the V4 CAD algorithm is now applied to PSCs as well. The CALIPSO project produces the Level 2 Polar Stratospheric Cloud product that uses the spatial and optical properties of these clouds to classify them according to type (Pitts et al., 2009). By definition a PSC is a cloud. In the CALIPSO PSC product, PSCs are classified by composition as a supercooled ternary solution (STS)

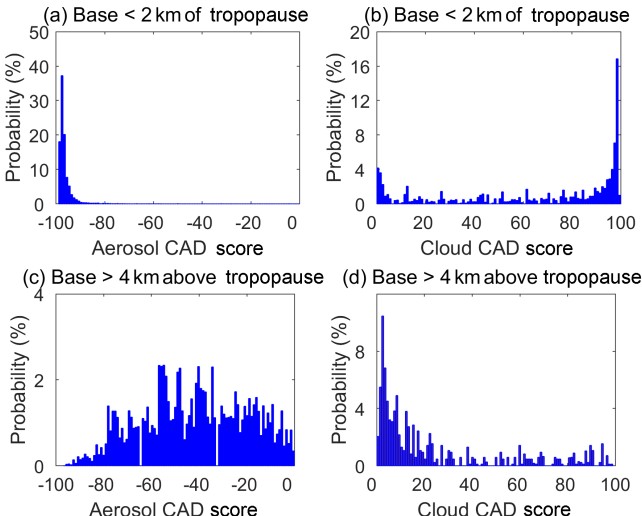

**Figure 19.** CAD scores of stratospheric layers detected at 5 km or coarser resolution observed during July through October 2008. The data in panels **(a)** and **(b)** are restricted to layers having base altitudes no more than 2 km above the tropopause, while the data in panels **(c)** and **(d)** have base altitudes that are more than 4 km above the tropopause.

of $HNO_3/H_2SO_4/H_2O$ CE7, Mix 1, Mix 2, or ice particles. Mix 1 and Mix 2 are PSC classes denoting lower and higher nitric acid trihydrate (NAT) number density and volume, respectively. Of these, STS may be thought of as closest to being a liquid aerosol particle. In this section, we assess the CAD classification of PSC layers by comparing V4 results with the classifications from the CALIPSO PSC-specific data product. Figure 20 shows a comparison of extensive PSC layers observed over Antarctica on 15 August 2008. Figure 20a shows results from the PSC product, with specific colors assigned to the different PSC classifications. Figure 20b shows the corresponding V4 CAD browse image for the same scene. Generally, there is a good correlation between the spatial distributions of STS in the left panels (red) and layers classified by V4 CAD as aerosols (red).

Figure 21 compares the spatial occurrence of stratospheric aerosol identified by the V4 CAD (left panels) and STS from the PSC product (right panels) for the months of June and January 2008 during the Antarctic and Arctic PSC seasons, respectively. There is a good correspondence between the locations of the peak concentration in latitude and altitude in both hemispheres. Despite the differences in spatial occurrence and the general uncertainty in applying cloud–aerosol terminology to PSCs, this level of correspondence is quite encouraging for the CAD performance.

Figure 22 shows the CAD scores assigned to polar stratospheric aerosols for January and June 2008, corresponding to the cases in Fig. 21. Above about 15 km the CAD scores are all very low (< 20) for these aerosol layers, similar to the clouds in Fig. 19. This is not unexpected since these parti-

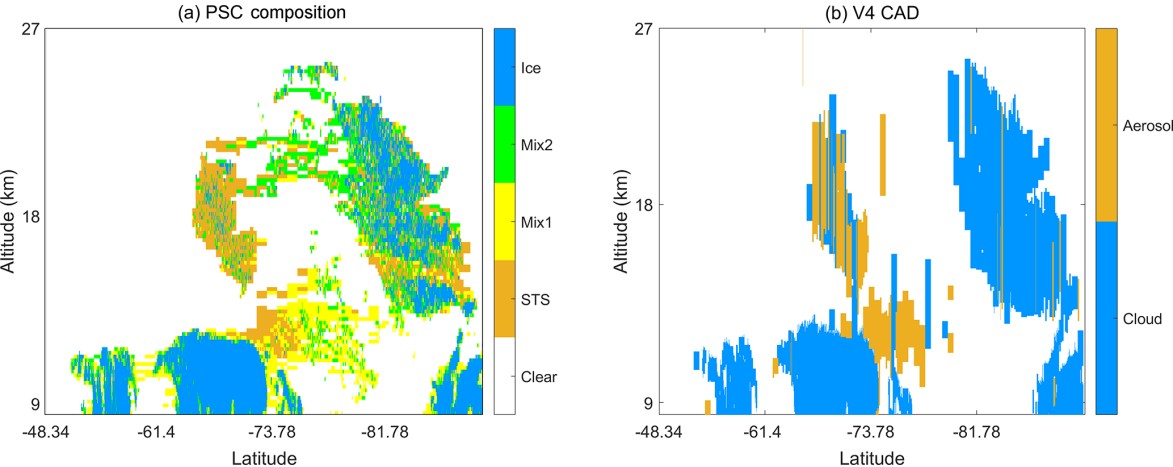

**Figure 20.** Spatial distributions of profiles of **(a)** STS from the CALIPSO PSC product and **(b)** stratospheric aerosol from the CALIPSO V4 CAD algorithm on 15 August 2008 over Antarctica for the granule 2008-08-15T15-25-28ZN.

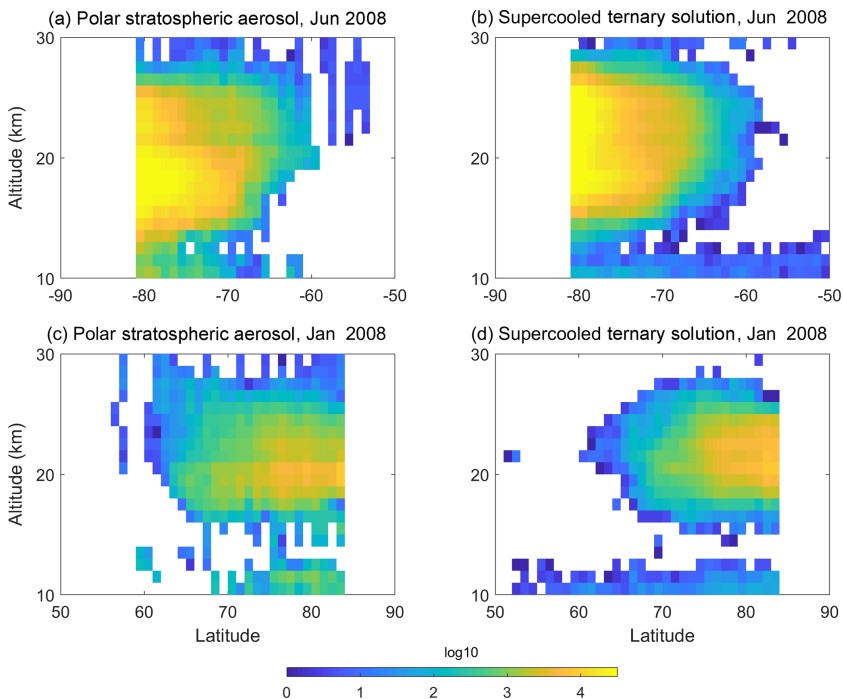

**Figure 21.** Comparison of the spatial distributions of the number of samples classified as stratospheric aerosols from the CALIPSO V4 VFM **(a, c)** and supercooled ternary solution (STS) from the CALIPSO PSC product **(b, d)** over the Antarctic in June 2008 **(a, b)** and over the Arctic in January 2008 **(c, d)**.

cles are in the process of becoming PSCs. However, noise in the 1064 nm data may also contribute to the classification uncertainties.

# 6  CAD for single-shot layers

Unlike the layers detected at 5 km and coarser resolutions (20 and 80 km), which are detected using the 532 nm mea-

surements, the single-shot layers at 333 m are detected by the CALIOP algorithm using the 1064 nm measurements between the surface and ∼ 8.2 km (Vaughan et al., 2009). These layers were classified a priori as clouds in all data releases prior to and including V3. This is because, pre-launch, aerosol layers were never expected to have the very high attenuated backscatter coefficients required to be detected at single-shot resolution. However, it has now been established that parts of extended dust layers that are exceptionally dense

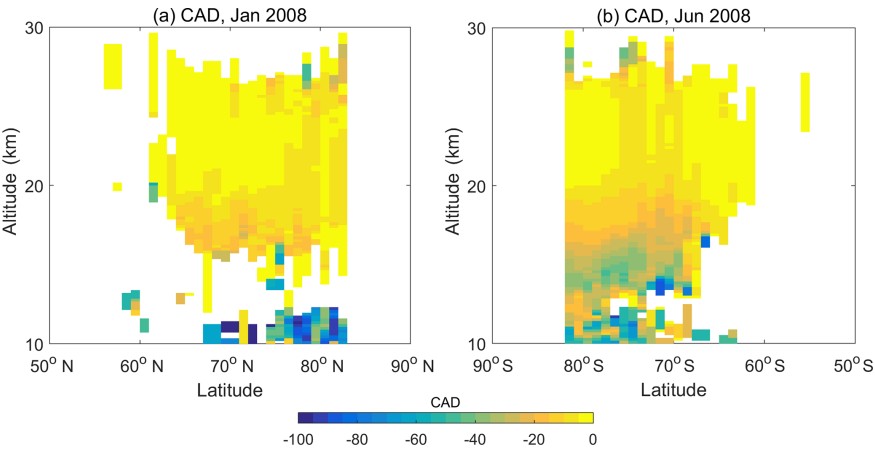

**Figure 22.** Spatial distributions of the CAD scores for polar stratospheric aerosols in **(a)** January and **(b)** June 2008.

can sometimes be legitimately detected at single-shot resolution. Thus, in V4, the CAD algorithm is applied to all layers detected at single-shot resolution. This section will assess the consequences of this change.

Figure 23 shows an example of layer classification of 333 m layers in V3 and V4, measured on 8 August 2008 when CALIPSO was passing over the Sahara desert. A strongly scattering layer can be seen between 10 and 15° N right over this desert area (pale green and orange colors in Fig. 23b). Embedded within the larger layer, a thick, dense layer with very high backscatter (in orange) can also be seen, which was detected at 333 m resolution (Fig. 23b) and classified as cloud (by default, not shown) in V3. However, as seen in Fig. 23c, almost all of this layer is now classified as aerosol in V4 (in orange). While clouds are occasionally embedded in extensive dust layers, in this instance over the Saharan desert the vertical extent and uniform backscattering within this thick 333 m layer strongly indicate that it is comprised of aerosols only. Findings such as this (which occur relatively frequently in the heart of the dust belt) demonstrate the usefulness of the V4 CAD algorithm even for the single-shot resolution layers.

While cloud layers at single-shot resolution have been observed all over the globe at various altitudes, dense aerosol layers amenable to detection at 333 m resolution are expected to occur mostly within extensive dust, marine, or smoke layers. Figure 24 shows the spatial distribution of the fraction of the 333 m layers that have been classified as aerosols by the CAD algorithm in V4 between 0 and 4 km during all months of 2008, for both daytime and nighttime data. As can be seen, the highest fractions of aerosol layers detected at 333 m resolution occur over the dust belt region from northeastern China to western Africa. Maximum fractions of $\sim 60\,\%$ and $\sim 40\,\%$ occur over the Sahara desert during the nighttime and daytime, respectively. Over all other areas, the aerosol fraction does not exceed $2\,\%$–$3\,\%$ of the total number of 333 m layers detected (i.e., both cloud and aerosol layers). Note that

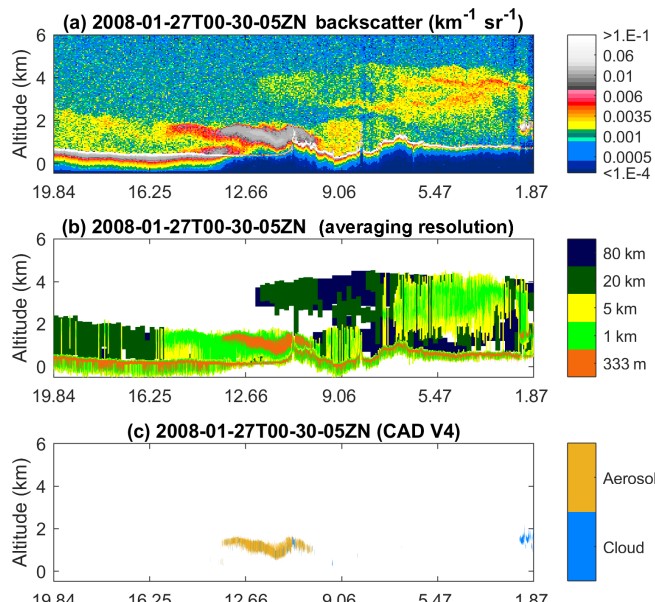

**Figure 23.** **(a)** V4 532 nm total attenuated backscatter coefficients and **(b)** horizontal averaging required for layer detection for a scene containing a strongly scattering aerosol layer observed on 27 January 2008 between latitudes of $\sim 13$ and $\sim 10°$ N. Cloud–aerosol classification of the V4 atmospheric layers detected at 333 m is shown in panel **(c)**. Note that the Earth's surface is also detected at the 333 m resolution, as seen in panel **(b)**, but these features are not plotted in panel **(c)**.

because single-shot detections were not included in the training set used for building the 5-D CAD PDFs, the V4 PDFs are not optimized for the classification of 333 m layers. As a result, there are cases in which extended, horizontally contiguous regions of 333 m layers are only partially classified as aerosols. These cases typically occur over arid regions, such as the Taklamakan Desert, and other regions of the globe where very high aerosol loading can be expected.

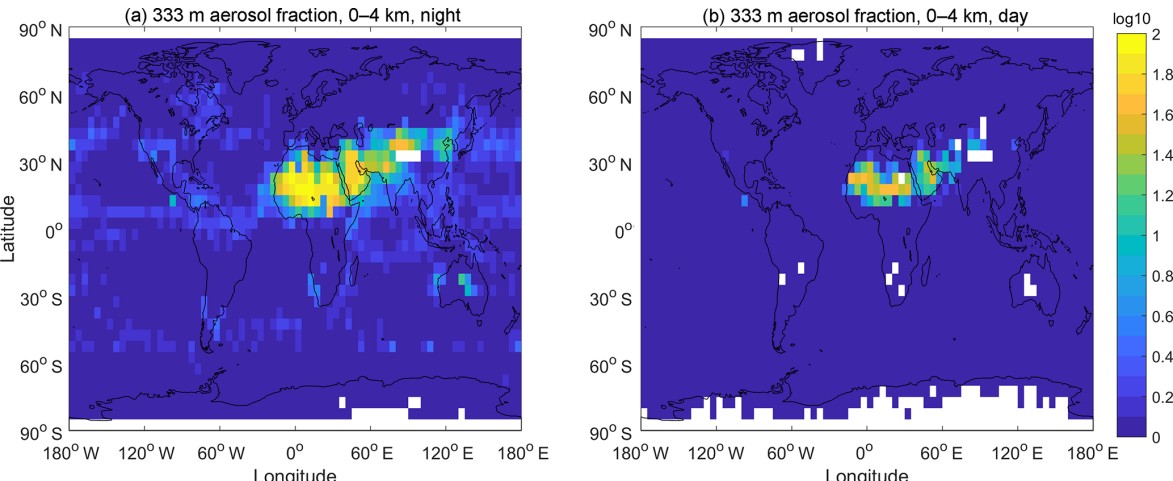

**Figure 24.** Spatial distribution of the fractional occurrence of the 333 m aerosol layers out of all layers detected at 333 m resolution between 0 and 4 km in 2008 for **(a)** nighttime and **(b)** daytime data.

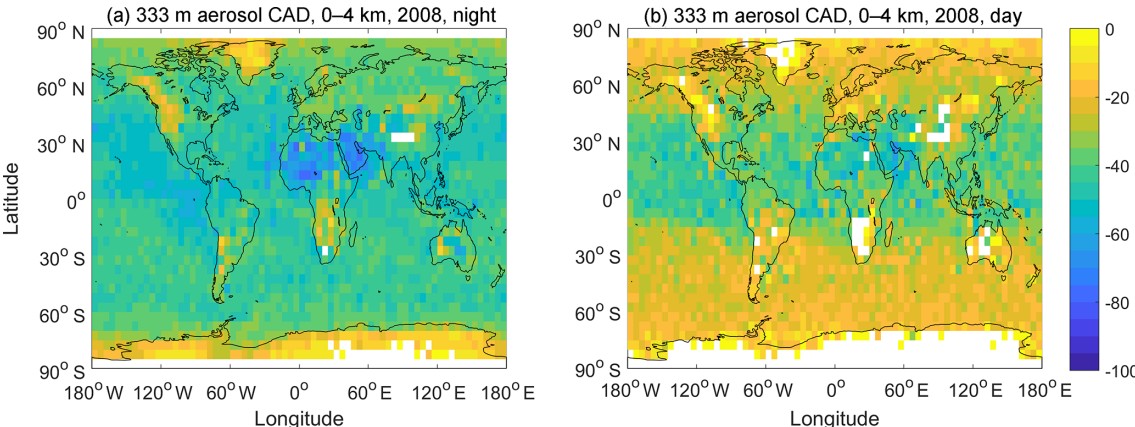

**Figure 25.** Spatial distributions ($5° \times 5°$) of the mean CAD scores for all 333 m aerosol layers detected during 2008 between 0 and 4 km; panel **(a)** shows nighttime means while panel **(b)** shows daytime means.

Figure 25 shows the spatial distribution of CAD scores for the 333 m aerosol layers detected between 0 and 4 km in the 2008 data. In general, the magnitude of the CAD scores is low ($< 50$) over most parts of the globe, with very low values ($< 20$) over Greenland and Antarctica. As noted above, the fractional occurrence of 333 m aerosols over these areas is also very low ($< 2\% – 3\%$), suggesting that these low CAD score magnitudes possibly reflect noise-related issues rather than being a systematic problem with the CAD algorithm. Note also the low CAD score magnitudes over the Taklimakan region. The single-shot resolution aerosol layers in this region are likely to be associated with dense dust layers, as seen over the Sahara, and thus higher CAD score magnitudes might have been expected for these layers. That we do not see these larger values is partly due to the fact that the Taklimakan region is farther north than the Sahara, and the occurrence of ice clouds is hence larger at the same altitudes due to the colder temperatures.

The CAD scores are comparatively robust over northern Africa (CAD score magnitudes $> 50$), where the fractional occurrence is also highest. Figure 25b shows the corresponding CAD scores over the daytime 333 m aerosol layers for 2008. Compared to the nighttime, the daytime layers had lower CAD score magnitudes everywhere, including the dust belt. Note that issues caused by the higher daytime noise and the lack of corrections for overlying attenuation can lead to imprecise CAD scores when the optical properties deviate significantly from those of 5 km layers used in the training sets.

Figure 26 shows the spatial distribution of the CAD scores for the 333 m cloud layers between 0 and 4 km for 2008 for nighttime (left) and daytime (right). Most of the cloud layers for both day and night have high CAD scores ($> 70$) as might be expected. However, over northern Africa, the CAD scores are relatively lower. As mentioned above, dense dust layers can be partially misclassified as cloud. These misclassified

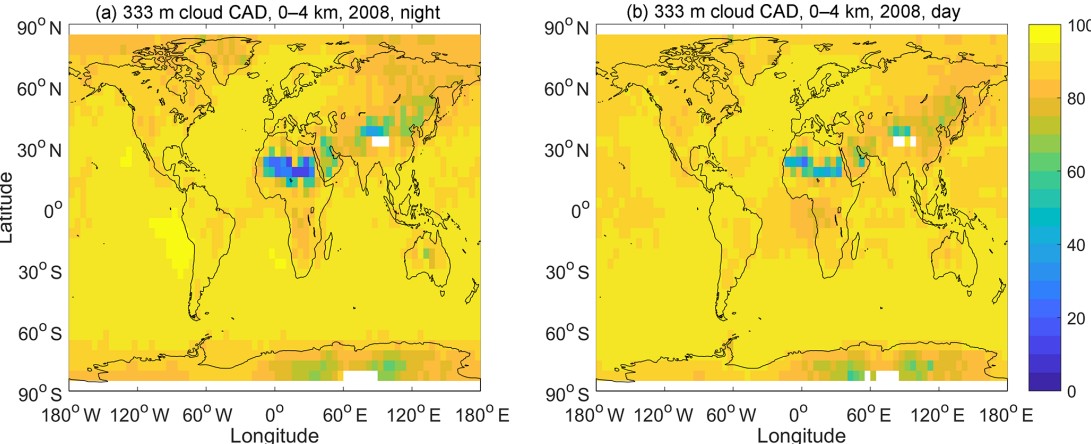

**Figure 26.** Spatial distributions of the CAD scores for all 333 m cloud layers detected between 0 and 4 km in 2008 for **(a)** nighttime and **(b)** daytime data.

clouds generally have low CAD scores and largely contribute to the low mean CAD score in this region where the cloud occurrence frequency is very low (see Fig. 15c and g).

## 7    Conclusions

In this paper, we have described the development and implementation of the probability distribution functions (PDFs) and post-processor algorithms used in the CALIOP version 4 (V4) level 2 cloud–aerosol discrimination (CAD) algorithm and provided preliminary performance evaluations via comparisons between the version 3 (V3) and V4 level 2 data products. Like the V3 PDFs, the V4 PDFs are constructed using five different spatial and optical properties: layer-integrated volume depolarization ratio ($\delta_v$), layer-integrated total attenuated color ratio ($\chi'$), layer-mean attenuated backscatter at 532 nm ($\langle\beta'_{532}\rangle$), latitude, and altitude. In addition, the new PDFs adopt finer spatial resolutions in the latitude and altitude dimensions, eliminate zero-scaling factors at high altitudes, and include high-altitude volcanic aerosols as part of the classification training data.

In contrast to previous versions, the V4 CAD algorithm is now applied to layers detected at all altitudes and at all horizontal resolutions. More particularly, using the CAD algorithm to evaluate layers detected at single-shot resolution has proven to significantly improve the classification of dense aerosols. The significantly improved calibration of the level 1 data, along with a more targeted development of the PDFs with higher latitude and altitude resolutions, leads to a more reliable and consistent separation between clouds and aerosols.

In evaluating the performance of the V4 CAD algorithm in the troposphere, we found that the classification of more than 95 % of clouds and daytime aerosols and about 89 % of nighttime aerosols remained unchanged between V3 and V4. Several of the systematic misclassifications observed in V3

(e.g., lofted Asian dust plumes being misidentified as cirrus clouds) have now been largely resolved. This should particularly benefit future studies of the transport of Asian dust to the Arctic in spring, the persistence of smoke at high altitudes, and volcanic aerosols injected into the stratosphere.

In the middle to upper troposphere, the V4 data products report a small increase in the fraction of optically thin cirrus clouds (i.e., cirrus fringes) that are misclassified as aerosol. This outcome is an unfortunate side effect of two highly beneficial improvements: the more accurate calibration of the 532 nm channel leads to more faint layers being detected, and the V4 aerosol PDFs have been redesigned to be more sensitive to the presence of depolarizing aerosol at high altitudes. The extent of these misclassifications is minimized using a newly developed cirrus fringe amelioration algorithm that uses a set of spatial proximity tests to evaluate and, as necessary, override the original aerosol classifications returned by the CAD PDF analyses. Additional safeguards are provided by a second post-PDF algorithm added to correct persistent misclassifications of dense water clouds lying beneath attenuating smoke plumes.

In the stratosphere, volcanic aerosol layers, polar stratospheric clouds (PSCs), and polar stratospheric aerosols (analogous to supercooled ternary solutions in standard PSC terminology) are now classified properly. However, as the altitude of the stratospheric layers increases above the tropopause, the decreasing signal-to-noise ratio of the measurements increases both the width of the PDFs and their degree of overlap, which in turn leads to low confidence in the classification of the layers (i.e., low CAD score magnitudes).

At single-shot resolution, the CAD algorithm again performs well, yielding the highest frequency of single-shot aerosol layers in the dust belt, which is consistent with geophysical expectations. The global distribution of CAD scores for single-shot detections also behaves as expected and desired. The CAD score magnitudes for single-shot aerosol lay-

ers are quite low in those places where very dense aerosol layers are not expected (e.g., over the southern oceans) and substantially higher over desert regions, where dense dust layers are relatively common.

Overall, in terms of the development of the PDFs and in the scope of their application, we find that the CALIPSO V4 lidar level 2 data products deliver substantial improvements in global cloud–aerosol discrimination relative to V3, and these more accurate classifications are expected to further improve the science results derived from CALIPSO measurements.

*Data availability.* This study made extensive use of the CALIPSO level 2 5 km merged layer product (Vaughan et al., 2018; NASA Langley Research Center Atmospheric Science Data Center; https://doi.org/10.5067/CALIOP/CALIPSO/LID_L2_05kmMLay-Standard-V4-10; last access: 10 July 2018), the CALIPSO level 2 5 km cloud profile product (Vaughan et al., 2018; NASA Langley Research Center Atmospheric Science Data Center; https://doi.org/10.5067/CALIOP/CALIPSO/LID_L2_05kmCPro-Standard-V4-10; last access: 10 July 2018), the CALIPSO level 2 5 km aerosol profile product (Vaughan et al., 2018; NASA Langley Research Center Atmospheric Science Data Center; https://doi.org/10.5067/CALIOP/CALIPSO/LID_L2_05kmAPro-Standard-V4-10; last access: 10 July 2018), and the CALIPSO level 2 vertical feature mask product (Vaughan et al., 2018; NASA Langley Research Center Atmospheric Science Data Center; https://doi.org/10.5067/CALIOP/CALIPSO/LID_L2_VFM-Standard-V4-10; last access: 10 July 2018). All CALIPSO lidar data products are also available from the AERIS/ICARE Data and Services Center (http://www.icare.univ-lille1.fr, AERIS/ICARE; last access: 10 July 2018).

## Appendix A: Improvement in feature detection due to improved data calibration

The CALIOP feature finder, referred to as the selective, iterated boundary location (SIBYL) algorithm, detects cloud and aerosol layers in the CALIOP backscatter signals (Vaughan et al., 2009). The SIBYL scheme embeds a generic profile scanning engine within an iterated, multi-resolution spatial averaging scheme. Each iteration of the profile scanning engine builds a range-varying detection threshold that scales automatically according to the magnitudes of the background noise and the expected molecular backscatter signal in the profile being examined. During execution of the scan, the threshold is further modified to account for the estimated attenuation of each feature encountered. By applying the multi-resolution averaging scheme, SIBYL reliably culls increasingly fainter features from increasingly coarser spatial averages of the same 80 km horizontal data segments. Due to the improvements in the V4 level 1 532 nm calibration procedures, which generally made the level 1 attenuated backscatter profile larger by 3 %–12 % or more (Kar et al., 2018; Getzewich et al., 2018), more faint features that went undetected in V3 are now detected in V4. As seen in Table A1, SIBYL's detection of clouds and aerosols increased by 6.8 TS2 % at night and 0.4 % during the day.

**Table A1.** Confusion matrices for V3 and V4 detection and classification derived from 1 year (2008) of the V3 and V4 cloud and aerosol profile products. The first three rows represent the V3 clear air, cloud, and aerosol categories, respectively. Similarly, the first three columns indicate the V4 clear air, cloud, and aerosol categories. The numbers in the diagonal elements are the fraction of each category for which the classification remained unchanged from V3 to V4. The numbers in the non-diagonal elements are the fractional changes from one category to the other relative to each V3 category. The last column is the fraction of each V3 category and the last row is the fraction of each V4 category relative to the total number of scenes. TS3

| **(a)** Night | V3 clear | V3 cloud | V3 aerosol | V3 total |
|---|---|---|---|---|
| V4 clear | 96.05 | 1.94 | 2.02 | 76.10 |
| V4 cloud | 2.31 | 93.17 | 4.52 | 16.71 |
| V4 aerosol | 6.58 | 10.35 | 83.07 | 7.20 |
| V4 total | 73.95 | 17.79 | 8.27 | 94.63 |
| **(b)** Day | V3 clear | V3 cloud | V3 aerosol | V3 total |
| V4 clear | 98.79 | 0.56 | 0.65 | 83.75 |
| V4 cloud | 2.42 | 93.38 | 4.20 | 12.23 |
| V4 aerosol | 7.55 | 4.22 | 88.23 | 4.02 |
| V4 total | 83.34 | 12.06 | 4.60 | 97.70 |

*Author contributions.* ZL developed the V4 cloud–aerosol discrimination (CAD) algorithm, derived the probability distribution functions, and spearheaded the comparative data analyses. JK created the initial outline of the paper. JK and SZ performed extensive evaluation of CAD performance in the stratosphere and troposphere, respectively. JT evaluated the classification of cirrus cloud fringes. MV edited the paper and managed the implementation of the algorithm. KPL, BG, and BM implemented the algorithm. MV, MA, JP, AO, PL, CT, and DW provided scientific guidance and helped design the data analyses and interpret the results.

*Competing interests.* The authors declare that they have no conflicts of interest. Coauthors Jacques Pelon and Charles Trepte are co-guest editors for the CALIPSO version 4 algorithms and data products special issue in *Atmospheric Measurements Techniques* but did not participate in any aspects of the editorial review of this paper.

*Special issue statement.* This article is part of the special issue "CALIPSO version 4 algorithms and data products". It is not associated with a conference.

*Acknowledgements.* This paper is dedicated to the memory of William H. (Bill) Hunt, pioneering lidar designer and inspirational mentor to the CALIOP algorithm development team. The authors also acknowledge the CALIOP engineers and technicians at NASA Langley Research Center and Ball Aerospace Technology Corporation. These men and women built a magnificent instrument that continues to perform superbly far, far beyond its original design lifetime. The two referees are thanked for their precious time and encouraging comments.

Edited by: James Campbell
Reviewed by: two anonymous referees

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

**Remarks from the language copy-editor**

**Remarks from the typesetter**