# Peer review of "Discriminating Between Clouds and Aerosols in the CALIOP Version 4.1 Data Products"

_Atmospheric Measurement Techniques, 2018_

## Referee Comment (RC1) · Anonymous Referee #1 · 19 Sep 2018

The manuscript "Discriminating Between Clouds and Aerosols in the CALIOP Version 4.1 Data Products" by Zhaoyan Liu et al. provides a detailed overview of how the new CALIPSO data product release separates clouds and aerosols. The extensive update to the development of the PDF for separating features based on the specified parameters were motivated well with previous version problems and overall improved science understanding since the previous data product release. The changes to both upstream development (e.g., calibration changes from V3 to V4) and potential downstream science (e.g., impact to science-related analysis) were also presented in the context of the improved data product for determining and interpreting CAD scores. Overall, the scientific significance/quality seemed in good shape. However, precision of the language could use some work to improve clarity, in particular for figure caption.

Below are some further technical comments:
- Figure 4: what are the diamond, red dots, square blue boxes? I'm guessing the ice cloud contours are at the same levels as the aerosols? Again guessing (k) is for all data?
- Figure 5: what are those contour for? Different colors from Figure 4.
- Page 17, line 3: "cirrus fringes with orange probability contour" alternatively, I think it would be easier to discern the figure as a reader by saying "cirrus fringes (left panel) and for adjacent cirrus layers (right panel). Orange probability contours for cirrus fringes and red probability contours are overlaid in both the left and right panel"
- Table 1: why is the occurrence frequencies outside the table description? Perhaps this is AMT style though.
- Page 24, line 7: is that CAD = 99 or is it CAD = +/-99? It is unclear if the discussion is for upper end confidence of clouds or both aerosols and clouds.
- Page 27, figure 12 caption: "The red curve in each panel is a column mean between 0 and 7 Km." as in the column mean of the percentage? What are the green lines in (d-f)? What is the thin red line in (f)?
- Page 32, line 20: "Table3" to "Table 3", space in between
- Page 34, Figure 15 caption: Caption refers to cloud fraction which typically ranges from 0 to 1 but then plots in log10(percentage), I was confused at first and I think this could be clarified in the caption.
- Page 35, Figure 16: label (b) clou $\Rightarrow$ cloud
- Page 37, line 9: word choice of vis-a-vis, considering changing to a more common phrase
- Page 43, Figure 23: What are the other features detected at single shot resolution that do not show up in 23c? I'm guessing it is the surface if it is not cloud/aerosol? Might be nice to point this out so readers do not focus on this when it is not the point of the figure.
- Page 45, Figure 25: So are those mean CAD scores and at what latitude-longitude resolution are those grid boxes?
- Page 46, line 8: word choice of gradation, considering changing to a more common phrase
- Page 47-48: so is the table confusion for all available data or for 2008 or some other time period?

---

## Referee Comment (RC2) · Anonymous Referee #2 · 23 Oct 2018

It is very clear that the authors have put a lot of effort in organizing and preparing the material for publication, and I believe it is ready for publication as is. The only fault is that it is a long paper, however I believe the length is justified given the importance of the work described here: CALIOP has collected more that 12 years of atmospheric data, and this paper describes the improvements to one of the core algorithms for the level 2 data processing. Down-stream CALIOP algorithms such as the extinction retrieval depend on accurate cloud/aerosol discrimination. This paper is a must read for anyone that uses CALIOP V4 data products.
* * *

---

## Author Comment (AC1) · 13 Nov 2018

**AMT-2018-190: Responses to Reviewers' comments**

Dear Editor

We thank you for taking care of this submission. We also appreciate the time that the referees have spent on the review of our lengthy manuscript and are grateful for their comments and insights. Below we have reproduced the referees' remarks (in black) and provided our responses to each (in blue). Any additional text added to the manuscript is shown in green.

Anonymous Referee #1

The manuscript "Discriminating Between Clouds and Aerosols in the CALIOP Version 4.1 Data Products" by Zhaoyan Liu et al. provides a detailed overview of how the new CALIPSO data product release separates clouds and aerosols. The extensive update to the development of the PDF for separating features based on the specified parameters were motivated well with previous version problems and overall improved science understanding since the previous data product release. The changes to both upstream development (e.g., calibration changes from V3 to V4) and potential downstream science (e.g., impact to science-related analysis) were also presented in the context of the improved data product for determining and interpreting CAD scores.

We thank you so much for the time and effort you have put into the review of this lengthy manuscript. We have revised our manuscript in accordance with your valuable comments. Please see our responses below following each of your comments/suggestions.

Overall, the scientific significance/quality seemed in good shape. However, precision of the language could use some work to improve clarity, in particular for figure caption. Below are some further technical comments:

- Figure 4: what are the diamond, red dots, square blue boxes? I'm guessing the ice cloud contours are at the same levels as the aerosols? Again guessing (k) is for all data?

  We have rewritten the caption to figure 4 as follows.

  Figure 4. Panels (a) – (j) show the joint distributions of V4 CALIOP measurements of $\chi'$ and $\langle \beta'_{532} \rangle$ (colored 2D distributions) for each of the PDF depolarization ratio intervals. These data were acquired during 2008 and June 2011 over a latitude band of 50°S-40°S and an altitude range of 12-16 km and were used to construct the V4 CAD PDFs that are applied within this same latitude–altitude region. Also shown in each panel are the derived PDFs for ice clouds (blue contours at three levels of 0.005, 0.05 and 0.5) and aerosols (red contour). Panel (k) aggregates all data in (a) – (j) and replots it in $\delta_v - \langle \beta'_{532} \rangle$ space, along with the $\langle \beta'_{532} \rangle_0$ values used to construct the PDFs for aerosols (red asterisks), ice clouds (blue squares) and water clouds (green diamonds).

- Figure 5: what are those contour for? Different colors from Figure 4.

  We replaced the plot in figure 5 with a new plot which uses the same colors as figure 4.

- Page 17, line 3: "cirrus fringes with orange probability contour" alternatively, I think it would be easier to discern the figure as a reader by saying "cirrus fringes (left panel) and for adjacent cirrus layers (right panel). Orange probability contours for cirrus fringes and red probability contours are overlaid in both the left and right panel"

We have modified the wording here as suggested by the referee.

- Table 1: why is the occurrence frequencies outside the table description? Perhaps this is AMT style though.

  We moved this text into the table caption.

- Page 24, line 7: is that CAD = 99 or is it CAD = +/-99? It is unclear if the discussion is for upper end confidence of clouds or both aerosols and clouds.

  Good point! The revised text reads as follows:

  The two most common CAD scores in V4 occur at CAD = –99 and CAD = 99.

- Page 27, figure 12 caption: "The red curve in each panel is a column mean between 0 and 7 Km." as in the column mean of the percentage? What are the green lines in (d-f)? What is the thin red line in (f)?

  Good catch! Thank you! These thin green and red lines should not be there. We have updated the plot. The red curve in each panel is a column average fraction (in %) between 0 and 7 km. To clarify our meaning, we have rewritten the caption for figure 12 as follows.

  Figure 12. Seasonal variations of (a) V3 aerosol fraction, (b) V4 aerosol fraction, (c) the fractions of V3 clouds changing to V4 aerosols, (d) V3 "hot cirrus", (e) V4 "hot cirrus", and (f) the fraction of V3 "hot cirrus" that changes to V4 aerosol in the Taklimakan region (35°N-45°N, 75°E-90°E). The fractions in (a), (c), (d) and (f) are relative to the total number of V3 clouds and aerosols whereas the fraction in (b) and (e) is relative to the V4 total. The red curve in each panel is the column average fraction (%) computed between 0 and 7 km. Additional information is provided in (c), where the blue curve shows the contributions from V3 "hot cirrus" and the magenta curve shows the contributions from all V3 ice clouds. The difference between the green and magenta curves quantifies the contributions of no confidence clouds, while the difference between the red and green curves quantifies the (very small) contributions from V3 water clouds. In total, 29.6 % of all V3 cloud-to-V4 aerosol changes were considered "hot cirrus" in V3. High confidence ice clouds, low confidence ice clouds, and water clouds contribute 47.1 %, 19.2 %, and 4.1 %, respectively.

- Page 32, line 20: "Table3" to "Table 3", space in between

  Done.

- Page 34, Figure 15 caption: Caption refers to cloud fraction which typically ranges from 0 to 1 but then plots in log10 (percentage), I was confused at first and I think this could be clarified in the caption.

  We have rewritten the caption for figure 15 as follows.

  Figure 15. Geographical distributions of V4 aerosol and cloud fractions (in log10 percentage) derived from one year (2008) of the CALIOP day (top two rows) and night (bottom two rows) profile products. Percentages are computed by dividing the number of 60 m × 5 km range bins classified as aerosol (panels a and e) or cloud (panels c and g) by the total number of range bins containing either aerosol or cloud. The corresponding

fractional changes from V3 cloud to V4 aerosol are shown in panels b and f.  Panels d and h show the fraction changes from V3 aerosol to V4 cloud.

- Page 35, Figure 16: label (b) clou ⇒ cloud

  Done.

- Page 37, line 9: word choice of vis-a-vis, considering changing to a more common phrase

  We replaced 'vis-à-vis' with 'when classifying', so that the sentence now reads as follows.

  Figure 18 shows two examples of V4 CAD performance when classifying stratospheric volcanic layers.

- Page 43, Figure 23: What are the other features detected at single shot resolution that do not show up in 23c? I'm guessing it is the surface if it is not cloud/aerosol? Might be nice to point this out so readers do not focus on this when it is not the point of the figure.

  Yes, agree. We have revised the caption for figure 23 as follows.

  Figure 23. (a) 532 nm total attenuated backscatter coefficients and (b) horizontal averaging required for layer detection for a scene containing a strongly-scattering aerosol layer observed on 27 January 2008 between latitudes of ~13°N and ~10°N. Cloud-aerosol classification of the V4 atmospheric layers detected at 333 m is shown in (c). Note that the Earth's surface is also detected at the 333 m resolution, as seen in (b), but these features are not plotted in (c).

- Page 45, Figure 25: So are those mean CAD scores and at what latitude-longitude resolution are those grid boxes?

  Yes, those are mean CAD scores and at 5°×5° resolution. We have modified the caption to clarify these points.

- Page 46, line 8: word choice of gradation, considering changing to a more common phrase

  We have changed "gradations" to "spatial resolutions".

- Page 47-48: so is the table confusion for all available data or for 2008 or some other time period?

  Good catch! Thanks! Yes, the table was derived for the year of 2008. To clarify, we added "from one year (2008)" in the caption.

Anonymous Referee #2

It is very clear that the authors have put a lot of effort in organizing and preparing the material for publication, and I believe it is ready for publication as is. The only fault is that it is a long paper, however I believe the length is justified given the importance of the work described here: CALIOP has collected more than 12 years of atmospheric data, and this paper describes the improvements to one of the core algorithms for the level 2 data processing. Down-stream CALIOP algorithms such as the extinction retrieval depend on accurate cloud/aerosol discrimination. This paper is a must read for anyone that uses CALIOP V4 data products.

Thank you so much for the encouraging words! We are especially gratified by your acknowledgement of the importance of the CAD algorithm, and your recognition of the time and effort our team has put into developing and testing the new CAD algorithm and preparing this manuscript. We sincerely hope that our many user communities will derive significant benefits from our version 4 data products.